# Cell assemblies at multiple time scales with arbitrary lag constellations

Eleonora Russo*, Daniel Durstewitz*

Department of Theoretical Neuroscience, Bernstein Center for Computational Neuroscience, Central Institute for Mental Health, Medical Faculty Mannheim, Heidelberg University, Mannheim, Germany

**Abstract** Hebb's idea of a cell assembly as the fundamental unit of neural information processing has dominated neuroscience like no other theoretical concept within the past 60 years. A range of different physiological phenomena, from precisely synchronized spiking to broadly simultaneous rate increases, has been subsumed under this term. Yet progress in this area is hampered by the lack of statistical tools that would enable to extract assemblies with arbitrary constellations of time lags, and at multiple temporal scales, partly due to the severe computational burden. Here we present such a unifying methodological and conceptual framework which detects assembly structure at many different time scales, levels of precision, and with arbitrary internal organization. Applying this methodology to multiple single unit recordings from various cortical areas, we find that there is no universal cortical coding scheme, but that assembly structure and precision significantly depends on the brain area recorded and ongoing task demands.

*For correspondence: eleonora. russo@zi-mannheim.de (ER); daniel.durstewitz@zi-mannheim. de (DD)

Competing interests: The authors declare that no competing interests exist.

## Introduction

Even more than six decades after its conception, Hebb's (1949) fundamental idea of a cell assembly continues to play a key role in our understanding of how neural physiology may link up to cognitive function. Loosely, a cell assembly refers to a group of neurons which, by functionally organizing into a temporally coherent set, come to represent mental or perceptual entities, thereby forming the basis of neural coding and computation (*Hebb, 1949*). However, the term lacks a stringent and universally accepted definition, and has been used to denote anything from the precise zero-phase-lag spike synchronization in a defined subset of neurons (*Abeles, 1991*; *Singer and Gray, 1995*; *Roelfsema et al., 1997*; *Diesmann et al., 1999*; *Harris et al., 2003*) to temporally coherent changes in average firing rates on larger time scales (*Goldman-Rakic, 1995*; *Durstewitz et al., 2000*). Often the term is meant to imply precise millisecond coordination of spike times for a 'volley' of activity which repeats at regular or irregular intervals in relation to specific perceptual or motor events (*Figure 1A, I*; e.g. [*Riehle et al., 1997*; *Roelfsema et al., 1997*; *Harris et al., 2003*; *Fries et al., 2007*]). Precise sequential patterns of spiking times (i.e., with time lags $\neq 0$) have been reported as well (*Figure 1A, II*), most commonly in the hippocampal formation where they may correspond to sequential orders of places (*Skaggs and McNaughton, 1996*; *Buzsáki and Draguhn, 2004*), in the visual cortex as a consequence of different activation levels (*König et al., 1995*), or as possibly generated through synfire-chain-like structures (*Abeles, 1991*; *Diesmann et al., 1999*). More generally, neurons may contribute several spikes in any order to a fixed spatio-temporal pattern (*Figure 1A, III*), as reported and linked to putative synaptic input motifs in vitro and in vivo (*Ikegaya et al., 2004*; *Yuste et al., 2005*). At a coarser temporal scale, neurons could fire with a specific temporal patterning to which each neuron may contribute 'bursts' of variable length (*Figure 1A, IV*). Such temporally ordered transitions among coherent firing rate patterns across sets of simultaneously recorded neurons have been described in different cognitive tasks and systems

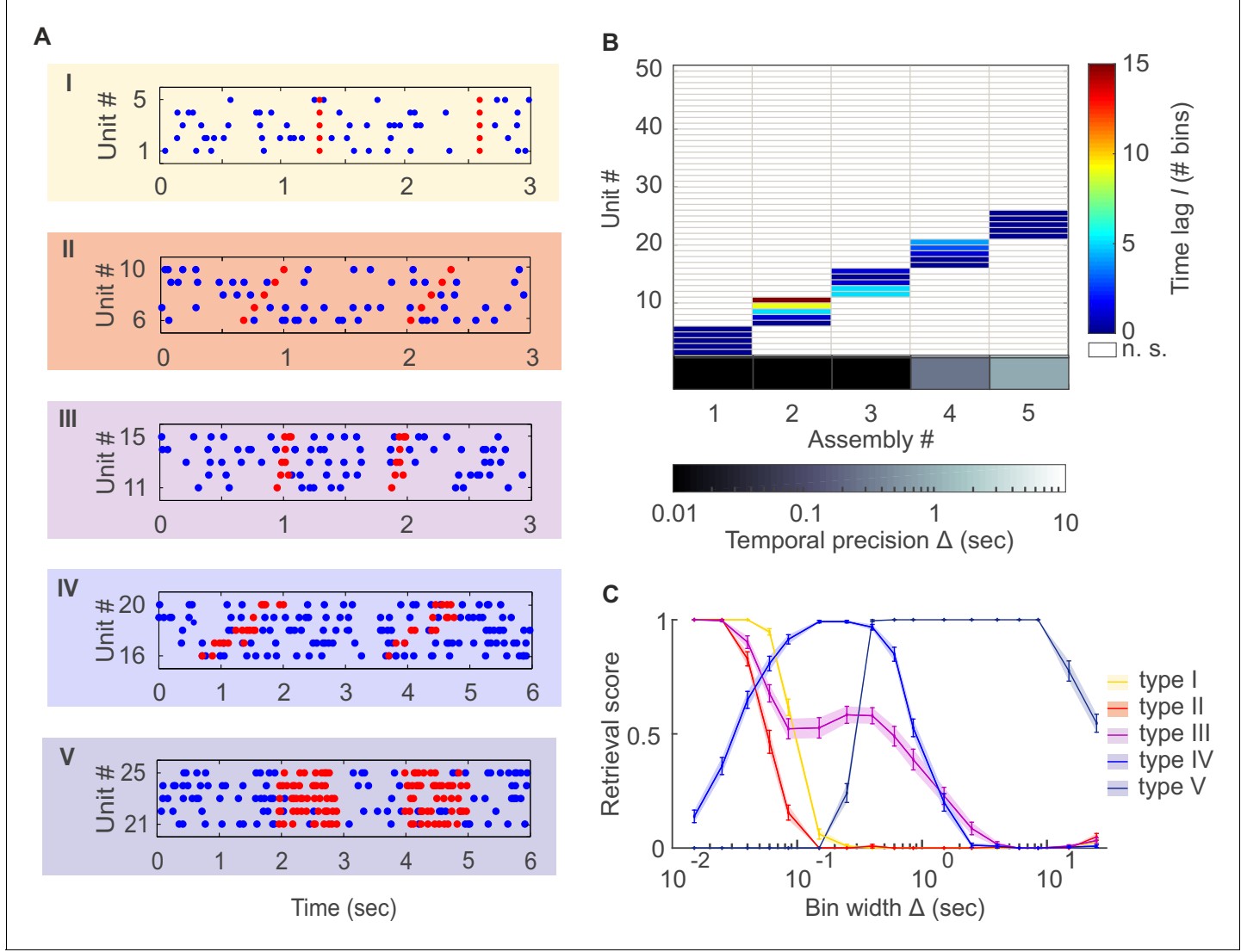

**Figure 1.** Detection of assemblies defined by different degrees of temporal precision, scale, and internal structure. (A) Different assembly types in simulated non-stationary spike trains: I –highly precise lag-0 synchronization; II – precise sequential pattern; III – precise spike-time pattern without clear sequential structure; IV – rate pattern with temporal structure; V – simultaneous rate increase. (B) Assembly-assignment matrix, showing how the 50 simulated units were grouped into assemblies, at which lags $l$ to the leading unit they were so (color-coded), and at which bin widths $\Delta$ the corresponding assemblies were detected (sorted along abscissa). (C) Assembly retrieval score (fraction of correctly assigned units) as a function of bin width for the different assembly types, averaged across 70 independent runs. Error bars = SEM.

The following figure supplement is available for figure 1:

**Figure supplement 1.** Dependence of synchronous pattern detection on reference lag.

(**Seidemann et al., 1996**; **Beggs and Plenz, 2003**; **Jones et al., 2007**; **Lapish et al., 2008**; **Durstewitz et al., 2010**). At a still broader temporal scale, sets of neurons jointly increasing their average rates for some period of time (**Figure 1A,V**), as during persistent activity in a working memory task, have also been linked to the cell assembly idea (**Durstewitz et al., 2000**).

There is indeed an ongoing, sometimes heated, controversy about the degree of temporal precision and coordination present in neural activity and its relevance for neural coding, partly based on empirical (**Shadlen and Movshon, 1999**; **London et al., 2010**), partly on statistical arguments (**Mokeichev et al., 2007**). Based on this discussion, it seems at present premature and limiting to focus on a single specific assembly concept, theoretical idea, or particular time-scale. Here we

develop a novel statistical approach for multi-cell recordings that treats the temporal scale, precision, and internal organization of coherent activity patterns as free parameters, to be determined from the data, and is thus open to a large family of possible assembly definitions (*Figure 1A*). By deriving a fast parametric test statistic for pairwise dependencies that automatically corrects for non-stationarity locally, computationally costly bootstrapping and sliding window analyses are avoided, reducing the computational burden by factors of 100–1000 (see *Materials and methods*). Thus, in combination with a computationally efficient agglomeration scheme which recursively combines units into larger sets based on significant relations detected in the previous step, considerable speed-ups are achieved. This in turn enables screening for assemblies at all possible lag constellations and temporal scales, not accomplished (to this extent) by previous algorithms to our knowledge (see *Materials and methods*). We then apply this methodology to examine in multiple single-unit (MSU) recordings from different cortical areas whether these employ a kind of universal temporal coding scheme, or whether and how the properties of the assembly code are adapted to the area-specific computations and task demands.

## Results

### Theoretical framework for assembly detection

From a statistical perspective, any of the assemblies from *Figure 1A* should reveal itself through recurring activity patterns in a set of simultaneously recorded spike trains, where a pattern can be any supra-chance constellation of unit activities with a specific distribution of time lags $l$ among them. The idea is to capture the multiple temporal scales introduced above through the width $\Delta$ used for binning the spike time series. We start from the relatively old notion of assessing the departure of the joint spike count distribution $p(A, B)$ of two units (or sets) $A$ and $B$ from independence (*Grün et al., 2002a*; *Pipa et al., 2008*). For two independent units with stationary spike trains, the joint distribution of spike occurrences at a specified time lag $l$ would factor into the single unit ('marginal') distributions, $p(A, B) = p(A)p(B)$. Assume each recorded spike time series has been converted into a series $\{c_t\}$ of spike counts of length $T$ at bin width $\Delta$, with $\#_A$ and $\#_B$ denoting the total numbers of spikes emitted by units $A$ and $B$, respectively. If $\Delta$ is small enough such that $c_t \in \{0, 1\}$ (binary counts), then, under the null hypothesis ($H_0$) of independence, the joint spike count $\#_{AB,l}$ at time lag $l$ follows a hypergeometric distribution with mean $\mu_{AB,l} = \#_A \#_B / (T - l)$ and variance $\sigma^2_{AB,l}$. If the binning is such that spike counts $c_t$ larger than one occur, the hypergeometric distribution is no longer directly applicable. We then split the series into several (mutually dependent) binary series (cf. Figure 6A) for which we obtain a joint mean and variance as derived in the *Materials and methods*.

The mean $\mu_{AB,l}$ and variance $\sigma^2_{AB,l}$ could, in principle, be used to check for deviation from the $H_0$ of independence at lag $l$, but in practice such a statistic would be corrupted by non-stationarities like (coupled) changes in the underlying firing rate (see *Materials and methods*, Figure 7, and *Appendix* on the importance of accounting for non-stationarity). Sliding window (*Grün et al., 2002b*) or bootstrap-based (*Fujisawa et al., 2008*; *Pipa et al., 2008*; *Picado-Muiño et al., 2013*) analyses have most commonly been used to deal with this issue, but come at the price of considerable data loss or computational burden. Here we suggest a simple remedy which corrects for non-stationarity locally by using the difference statistic $\#_{ABBA,l} = \#_{AB,l} - \#_{AB,-l}$ (see *Materials and methods*, Figure 6B). The idea is that this way non-stationarities in firing rates would cancel out locally, on a comparatively fine time scale ($\approx l\Delta$), since they would affect $\#_{AB,l}$ and $\#_{AB,-l}$ alike (for assessment of synchronous spiking, we use $\#_{ABBA,0} = \#_{AB,0} - \#_{AB,l^*}$, with $l^* = -2$; see sect. '*Choice of reference (correction) lag*' for the motivation of this particular choice and a more general discussion of the reference statistics chosen). The statistic $Q_l \equiv \#_{ABBA,l}^2 / \hat{\sigma}^2_{ABBA,l}$ finally is approximately $F$-distributed and can be used for fast parametric assessment of the $H_0$ (see *Materials and methods* and Figure 7; *Figure 7—figure supplements 1* and *2*, for derivation and empirical confirmation using non-stationary synthetic data).

Having derived a fast, non-stationarity-corrected parametric test statistic for assessing the independence of pairs, we designed an agglomerative, heuristic clustering algorithm for fusing significant pairs into higher-order assemblies (see *Figure 6—figure supplement 1* and *Materials and methods* for full derivation and pseudo-code). In essence, at each agglomeration step the algorithm treats each set of units fused in an earlier step just like a single unit with activation times defined

through one of its member units. This allows for the same pair-wise test procedure on sets of units as defined for single units above, while at the same time effectively testing for higher-order dependencies based on the joint (set) distributions (see *Materials and methods*). Each pair is tested at all possible lags $l \in \{-l_{max} \ldots l_{max}\}$ (with $l_{max}$ provided by the user), which is a reasonably fast process given the parametric evaluation introduced above. Should a pair of unit-sets prove significant at several lags $l$ at any step, only the one associated with the minimum *p*-value is retained. The recursive set-fusing scheme stops if no more significant relationships among agglomerated sets and single units are detected. All subsets nested within larger sets are then discarded. This whole procedure is repeated for a set of user-provided bin widths $\Delta \in \{\Delta_{min} \ldots \Delta_{max}\}$. For each formed assembly, the width $\Delta^*$ associated with the lowest *p*-value may then be defined as its characteristic temporal precision. All tests are performed at a user-specified, strictly Bonferroni-corrected $\alpha$-level (always set to 0.05 here; see *Materials and methods* for details).

## Performance evaluation on simulated data

The agglomerative scheme described above is a fast heuristic proxy, similar in spirit to the apriori algorithm in machine learning (*Hastie et al., 2009*; *Picado-Muiño et al., 2013*), for evaluation of all possible unit and lag combinations. To illustrate and evaluate its performance, synthetic data with known ground truth were created. Cell assembly structures with the different levels of temporal precision and internal organization (i.e., lag distributions) as shown in *Figure 1A* were simultaneously embedded within inhomogenous (i.e., non-stationary) Poisson spike trains, with a mean rate following an auto-regressive process (see *Materials and methods*). The assembly-assignment matrix in *Figure 1B* demonstrates that all five different types of assemblies (and only these, no false detections) were correctly identified with their associated temporal precision and lag distributions. *Figure 1C* illustrates the quality of 'assembly retrieval' (measured as fraction of assembly units correctly assigned) as a function of bin width $\Delta$: As expected, the retrieval quality steeply declines for the temporally precise assemblies as the bin width increases (types I and II), while it rises up to the appropriate temporal scale for the more broadly defined assemblies (types IV and V). For assembly-type III, defined by precise temporal relationships, yet extended across time without strictly sequential structure, both these time scales are revealed (leading to the local peak at ~300 ms). Also note that the correlated rate increases which define assemblies of types IV and V naturally can be discovered already at lower bin widths than the one which corresponds to the temporal extent of the whole pattern. We also investigated more systematically (*Figure 2*, see also *Materials and methods*) how assembly retrieval varies as a function of sample size and potential spike sorting errors. Assembly detection starts to significantly degrade only when their relative contribution to the spike series drops below ~4% (*Figure 2A*), or when more than ~30% of all spike times were corrupted by spike sorting errors (*Figure 2B*). More importantly, across a whole range of sample sizes, spike assignment errors, and assembly structures tested, the fraction of units *falsely* ascribed to any one assembly stayed uniformly low at about 0.5% (*Figure 2C,D*), indicating that our procedure is quite conservative and rarely returns false positives in the simulated scenarios.

## Area- and task-specific assembly configurations and time scales

We next examined assembly structure in different brain regions from which multiple single-unit recordings were obtained in previously published experiments, including the rat anterior cingulate cortex (ACC; [*Hyman et al., 2012*; *Hyman et al., 2013*]), hippocampal CA1 region, and entorhinal cortex (EC, [*Pastalkova et al., 2008*; *Mizuseki et al., 2009*, *2013*]) (see *Materials and methods* for further specification). *Figure 3A* presents the assembly-assignment matrix from one of the ACC data sets. Detected assemblies span a large range of temporal precisions, from ~10 ms to about 1.5 s, with a variety of lag distributions, and are composed of about 10% (ACC) to 16% (CA1, EC) of the recorded neurons. Note that different from the clear-cut hypothetical examples (*Figure 1B*) which were strictly disjoint by design, many of the experimentally recorded assemblies partially overlap (i.e., share units; see also *Materials and methods*). *Figure 3B* also gives specific examples of assemblies with relatively high (top) and with lower (bottom) temporal precision. Finally, many of the unraveled assemblies were highly selective for specific task events as illustrated in *Figure 3C*, *Figure 3— figure supplement 1*.

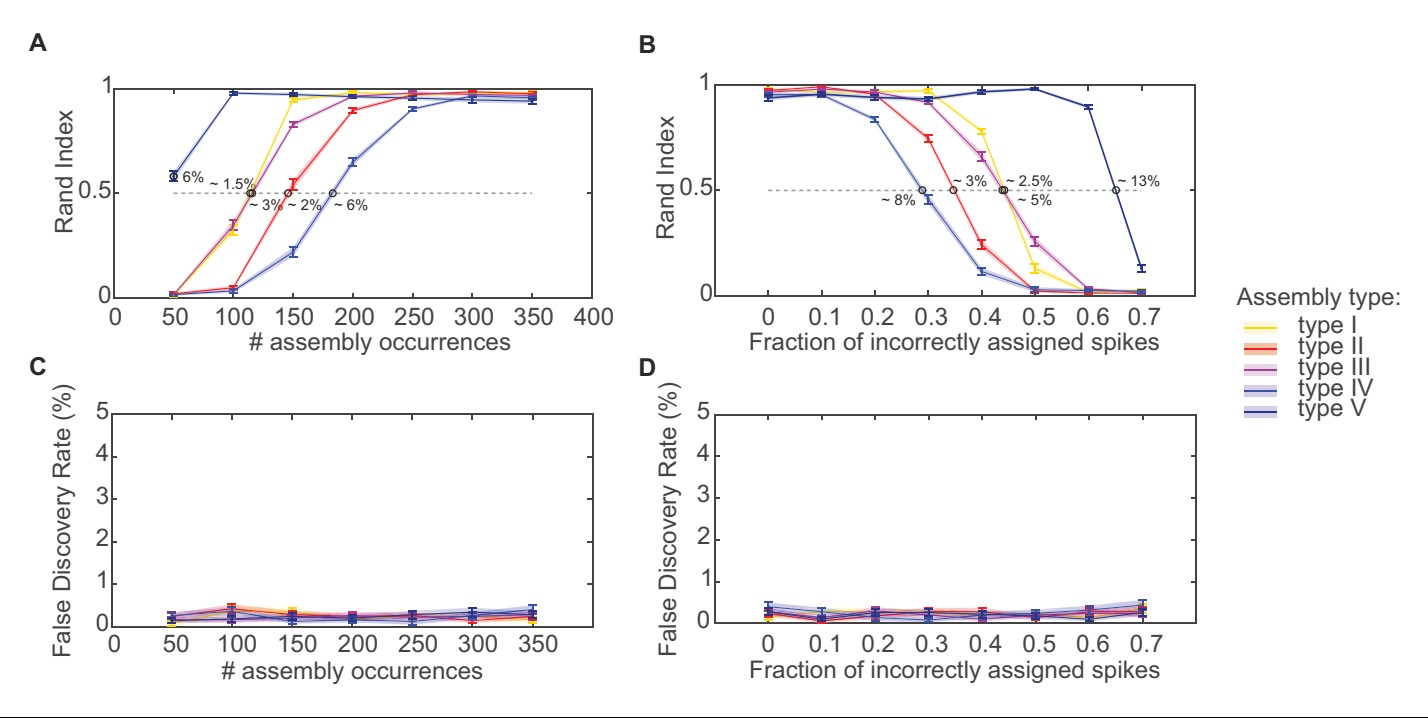

**Figure 2.** Performance evaluation of assembly detection algorithm. (**A**) Rand index $R(r, s)$, *Equation 14*, as a function of the total number of occurrences of the assembly pattern in spike time series of length $T$=1400 s, averaged across 50 independent runs, for all five types of assemblies from *Figure 1* (as indicated in the inset legend). Percentages at the half-width points of the $R(r, s)$-curves indicate the proportion of spikes in the time series contributed by the assembly structures at these points. (**B**) Rand index for all assembly types as a function of the fraction of incorrectly assigned spikes ('sorting errors'). (**C**) Fraction of units incorrectly assigned to an assembly across a range of assembly occurrence rates. (**D**) Fraction of units incorrectly assigned to an assembly as a function of the fraction of misattributed spikes. Error bars = SEM.

A specific question one might ask is whether different brain areas host different types of assembly structures, and how these may depend on the behavioral task. These aspects are quantified in *Figure 4A* by plotting the distribution of all significant unit pairs as a function of bin width $\Delta$ and time lag $l$. Several features are noteworthy here: First, the joint $(\Delta, l)$-distribution changes dramatically as the animals move from unstructured, completely self-paced, little demanding environmental exploration (*Figure 4A*, left column) to a highly structured, demanding delayed alternation task (*Figure 4A*, right column). During the latter, a much larger number and richer repertoire of assembly structures turned out, as also indicated by the 'marginal' distributions of significant unit pairs across time scales $\Delta$ in *Figure 4B*. However, secondly, while these changes in ACC and CA1 were mostly focused on the larger timescales, in EC they appeared to run across all timescales, yet were overall less dramatic than in CA1 (*Figure 4A*, bottom; *Figure 4B*, bottom). Third, assemblies remarkably differ among the three brain regions in the range of temporal scales they occupy: While EC mainly harbors fine temporal structure with a precision of about 15–50 ms, in ACC broad rate change patterns in the 120–700 ms range appear to dominate (*Figure 4B*). CA1, in contrast, expresses both temporal scales, or in fact a wide spectrum from about 30–1500 ms, of which the broader scales (>100 ms) only surface during delayed alternation. Overall, a much larger number of units engaged in any one assembly in CA1 (>90%) than in ACC (30–50%; *Figure 4B*). These observations indicate that the temporal composition and precision of assembly activity strongly depends on both the brain area and the behavioral setting.

On closer inspection, some of the temporally more precise 30–50 ms assemblies in CA1 were found to code for specific place fields ('place assemblies') in the rat's environment (*Figure 5A*, *Figure 5—figure supplement 1*). These assemblies mainly consisted of synchronous (lag-0) spiking units (*Figure 4A*). Meanwhile, the more broadly tuned assemblies in CA1 tended to code for temporally extended events which often appeared to have a specific behavioral meaning in the task context:

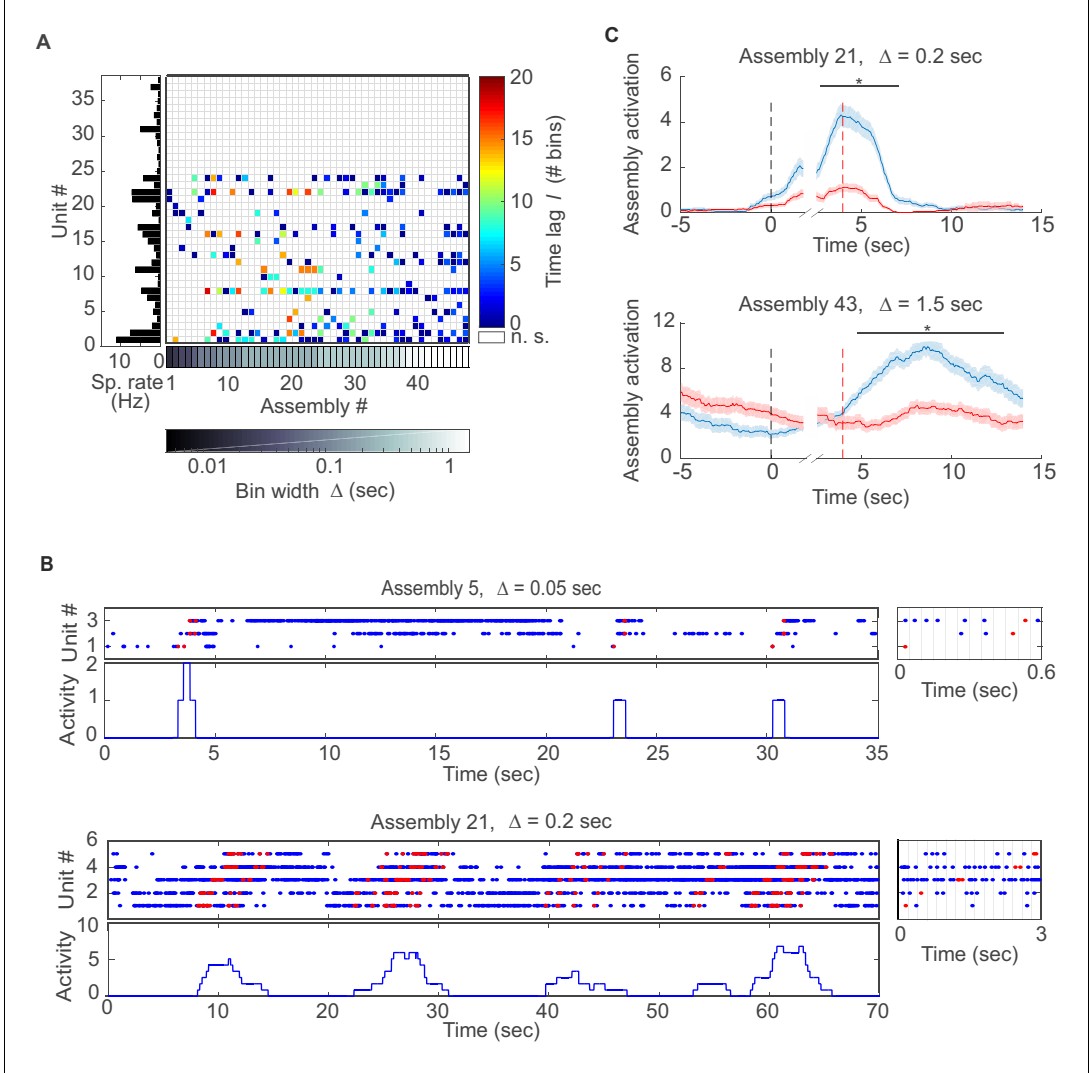

**Figure 3.** Assemblies in recordings from anterior cingulate cortex (ACC) during delayed alternation. (A) Assembly-assignment matrix for one ACC data set, with the average firing rate of units indicated on the left. (B) Examples of detected assembly patterns at relatively precise (top; 50 ms) and broader (bottom; 200 ms) time scales. Insets on the right zoom in on detected assemblies with optimal binning Δ indicated by vertical lines. See *Material and methods* for computation of assembly activation scores ('activity') as shown in the lower panels. (C) Two examples of selective assembly activity discriminating between left (blue curves, n=39) and right (red curves, n=34) lever presses during actual lever press (top) or during delay (bottom). Times of lever press and nose poke are indicated by vertical red and black dashed lines, respectively. Periods of significant differentiation indicated by black bars above curves (two-tailed, paired t-test, *p<0.05, Bonferroni-corrected for number of bins tested, *Figure 3—source data 1*). Shaded areas = SEM.

The following source data and figure supplement are available for figure 3:

**Source data 1.** Assembly activation in different trials relative to left/right lever press for assemblies n. 21 and n. 43.
**Figure supplement 1.** Examples of assembly activation in a delayed alternation task.

For instance, these assemblies may become active during the reward event irrespective of its spatial location (*Figure 5B*), or for the whole correct choice path after a behavioral decision was made (*Figure 5D*). These temporally broader assemblies commonly also followed a more sequential (lag≠0) layout (*Figure 4A*). Interestingly, the single cells constituting CA1 assemblies did not necessarily share the same place preference with their 'parent' assembly (*Figure 5—figure supplement 1*). Similar as in CA1, broader assemblies in ACC were tuned to specific task phases and events

(lever presses, delays, stimulus conditions) and reflected the task's sequential structure (*Figure 3C*, *Figure 3—figure supplement 1*).

## Discussion

Here we introduced a novel theoretical and statistical framework, based on fast parametric testing and computationally efficient agglomerative algorithms, which detects assembly structure at many different temporal scales, and with arbitrary internal organization, while at the same time accounting for non-stationarity on a fine time scale. This enables to readdress fundamental questions about the temporal structure and nature of neural representations in a largely unbiased way. One potential caveat to be noted here, however, is that the particular choice of reference bin for removing non-stationarity still entails a (mild) assumption about structure: For the present choice of pairing $\#_{AB,l}$ with its time reverse, $\#_{AB,-l}$, it is that temporal dependencies are essentially directed (except for the synchronous case), i.e. do not simultaneously, within a given pair of time series, occur with *exactly* the same time lag in both directions. If in doubt about this, however, analyses may be repeated with another (not too large) reference lag (see sect. '*Choice of reference (correction) lag*' for a discussion of alternative choices and factors to consider, including differences in sensitivity to non-stationarity implied by different lag spans covered by the test statistic). Testing other reference lags may also help with types of non-stationarity that may violate the conditions defined below *Equation 6*.

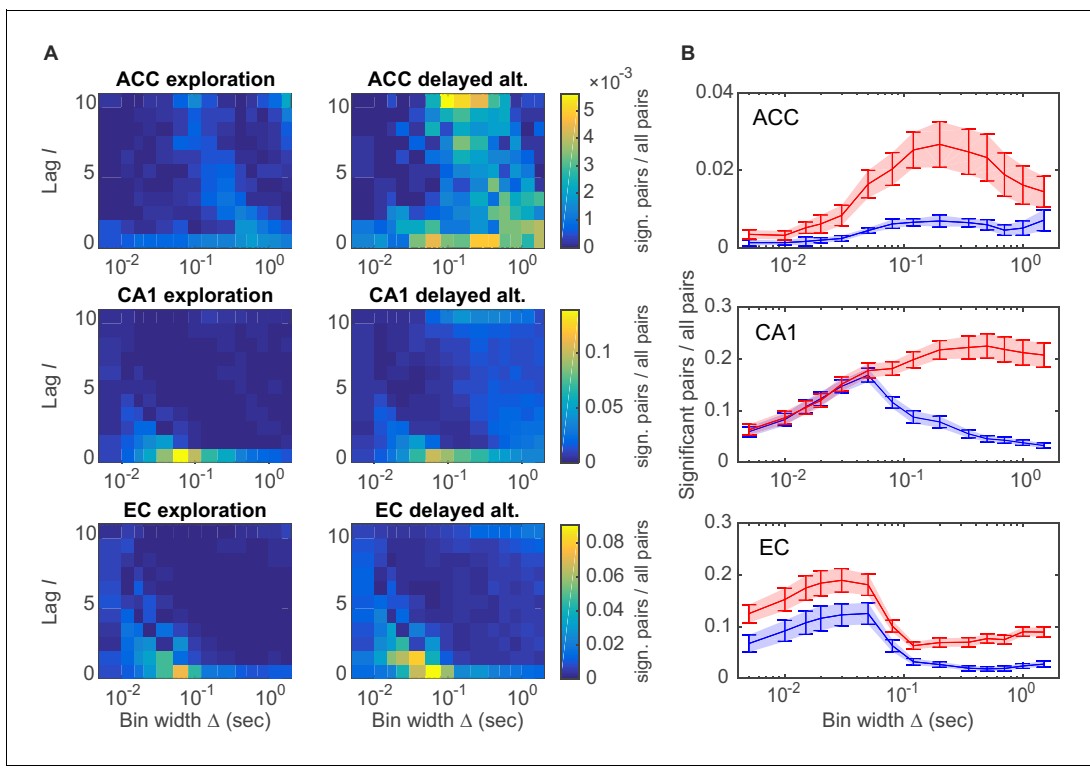

**Figure 4.** Assembly structure in different brain areas and behavioral tasks. (**A**) Relative frequency histograms (color-coded) of all significant unit pairs, pooled across all detected assemblies, as a function of characteristic time scale $\Delta \in \{0.015 \ldots 1.5\}$s and lag $l \in \{-10 \ldots 10\}$, for the anterior cingulate cortex (ACC, top), CA1 region (center), and entorhinal cortex (EC, bottom) during environmental exploration (left; total numbers of pairs tested: 9316 [ACC], 7597 [CA1], 5024 [EC]) and delayed alternation (right; total numbers of pairs tested: 4090 [ACC], 9847 [CA1], 4183 [EC]). Note that the aggregation at larger lags for ACC is partly due to the fact that the algorithm considered lags up to $l_{max} = 10$, such that significant pairs with optimal lag $l>10$ have been assigned to $l_{max}$. All tests are Bonferroni-corrected for numbers of pairs and lags tested. (**B**) Marginal distributions of significant assembly-unit pairs across temporal scales $\Delta$ for ACC (top), CA1 (center), and EC (bottom). Blue curves = environmental exploration. Red curves = delayed alternation. Shaded areas = SEM.

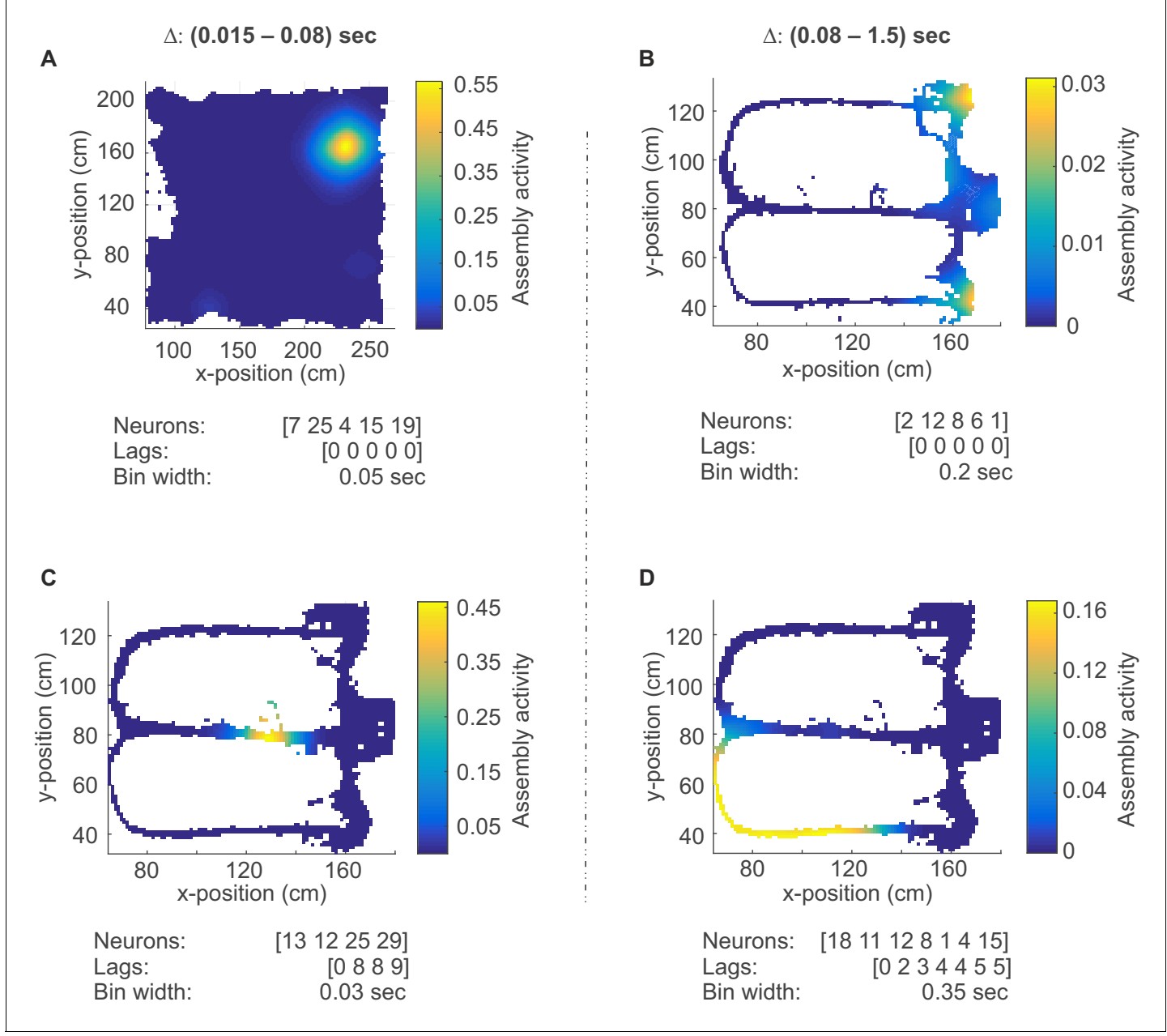

**Figure 5.** Assembly coding at different time scales in CA1. Color-coded activity maps for four CA1 assemblies during an environmental exploration task (A) and a delayed alternation task (B, C, D). Below x-axis in each panel: Identities of neurons assigned to the assembly, associated time lags within an assembly, and temporal scale of assembly.

The following figure supplement is available for figure 5:

**Figure supplement 1.** Single-unit composition of cell assemblies.

Illustrating this methodology on multiple single-unit recordings from ACC, CA1, and EC, it appeared as if the temporal structure and precision of the revealed assemblies were closely related to the computations performed by these brain areas: While the CA1 region processes precise spatial (*O'Keeffee and Nadel, 1978*; *Harris et al., 2003*; *Diba and Buzsáki, 2007*) and temporal (*Eichenbaum, 2014*) environmental structure, the ACC is much less concerned with finely-granulated details of the spatial world (*Hyman et al., 2012*). Rather, activity in ACC reflects behavioral organization,

behavioral monitoring, overall context, and task structure, processes which typically unfold on much slower temporal scales (*Lapish et al., 2008*; *Hyman et al., 2012*). Likewise, in addition to spatial coding, the hippocampal CA1 region has also been reported to represent aspects of higher-order decision making, like paths to a defined goal state or choice outcomes (*Lisman and Redish, 2009*; *Buzsáki, 2015*). These capacities may become relevant only when an animal is transferred from unstructured environmental exploration to a task which involves clearly defined goal states, reward-related choices, and possibly time delays between them. Consequently, sequential organization of assemblies at broader time scales was much more often observed in the latter than in the former task context.

Numerous other statistical procedures for detecting assemblies or sequential patterns have been proposed previously (*Grün et al., 2002a*; *Grün et al., 2002b*; *Pipa et al., 2008*; *Torre et al., 2016a*), but most of these adhere to one or the other theoretical conceptualization of a cell assembly (cf. *Figure 1A*), or become computationally impractical for larger cell numbers or multiple lags (see *Appendix* for further discussion of both more recent and more 'traditional', cross-correlation-based, approaches). Also, none of these, to our knowledge, combines all of the features presented here. The statistical tools developed here may allow readdressing questions about the nature of neural coding in different brain areas, without requiring the researcher to commit to any particular assembly concept or theoretical framework a priori. Indeed, we observed that there may be not just one type of cortical assembly code, but that the temporal precision, scale, and sequential composition with which cortical neurons organize into coherent patterns strongly depends on the brain area and task context investigated. We further note that our methods are not specific to the neuroscientific domain, but could be used more widely in other scientific areas to detect structure at multiple temporal scales in multivariate event count series.

## Materials and methods

### Statistical test for pairwise interaction

Assume we have recorded $N$ spike trains, each divided into $T$ bins of equal bin width $\Delta$, resulting in a spike count series $\{c_{K,t}\}$, $t = 1 \ldots T$, for each recorded unit $K$. Bin width $\Delta$ sets the temporal precision at which unit interactions are to be detected. We would like to test for a range of time lags $l \in \{-l_{max} \ldots l_{max}\}$ whether the joint spike count $\#_{AB,l}$ of units $A$ and $B$ at lag $l$ significantly exceeds of what would have been expected under the null hypothesis ($H_0$) of independence of the two spiking processes. For clarity, note that $\#_{AB,l}$ is computed by counting the number of times we have a spike in unit $A$ and a corresponding spike in unit $B$ $l$ time bins later. From the range of all considered lags, we select the one which corresponds to the highest count $\#_{AB,\bar{l}}$, i.e. $\bar{l} \equiv \operatorname{argmax}_l(\#_{AB,l})$. For deriving the proper distributional assumptions under the $H_0$, spike count series $\{c_{K,t}\}$ are often thresholded (*Grün et al., 2002a*; *Humphries, 2011*; *Shimazaki et al., 2012*; *Picado-Muiño et al., 2013*) such that binary {0,1}-series are obtained, presumably partially since multivariate extensions of the binomial or hypergeometric distribution are not yet commonplace (see *Teugels, 1990*; *Dai et al., 2013*). Especially for larger bin widths $\Delta$ this implies a serious loss of information, however. Therefore, we sought a different approach to the problem that makes use of the full spike counts, based on the first two moments of a multivariate hypergeometric distribution. Instead of thresholding, we split each spike count series into a set of $\alpha = 1 \ldots M_K$ binary processes as indicated in *Figure 6A*, where $M_k \equiv max(c_{K,t})$ is the maximum spike count observed for unit $K$ for the specified bin width. The first binary process is defined by having a '1' in all time bins for which spike count $c_{K,t} \geq 1$ (and '0' otherwise), the second process by having a '1' only in those time bins for which $c_{K,t} \geq 2$, and so on. For any two units $A$ and $B$, defining $M \equiv min(M_A, M_B)$, the total joint count $\#_{AB,\bar{l}}$ at selected lag $\bar{l}$ is now simply given by the sum of joint counts $\#^{\alpha}_{AB,\bar{l}}$ across all pairs of binary subprocesses $\alpha = 1 \ldots M$,

$$\#_{AB,\bar{l}} = \sum_{\alpha=1}^{M} \#^{\alpha}_{AB,\bar{l}}. \tag{1}$$

Since each of the $M$ subprocesses is binary, each number $\#^{\alpha}_{AB,\bar{l}}$ follows a hypergeometric distribution under the $H_0$ (since marginal counts $\#^{\alpha}_A$ and $\#^{\alpha}_B$ are fixed by the observed data, a binomial

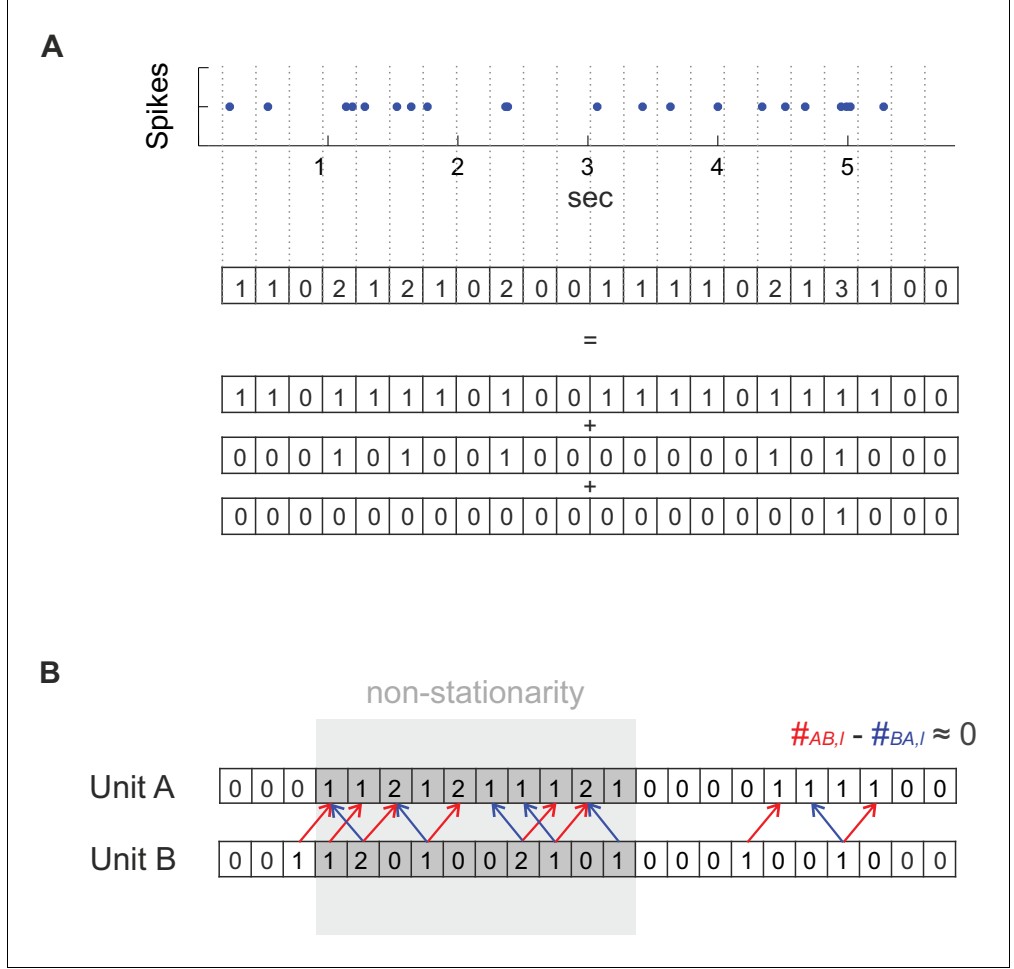

**Figure 6.** Method details. (**A**) For deriving a statistical test that works with any temporal bin width the spike count series were separated into an overlay of several (dependent) binary sub-processes. See *Materials and methods* for further explanation. (**B**) Dealing with non-stationarity in the spike trains. In the case of non-stationarity in the form of a common rate increase in two units A and B (highlighted in gray), some spike co-occurrences caused by the rate increase might be incorrectly attributed to coupled activity (mutual dependence) at the finer timescale (bin width) at which coupling is investigated (at a lag of one in the illustrated example), even if there is not really any such coupling as assumed in this example. This corruption by non-stationarity may be removed by considering the difference count $\#_{AB} - \#_{BA}$, in which spurious excess coincidences in one direction ($\#_{AB}$: red arrows) would cancel out with those in the reverse direction ($\#_{BA}$: blue arrows). It is important to note that if, on the other hand, the rate increase is on the timescale of interest, as it is the case for the 'rate assemblies' of type IV or V in *Figure 1*), subtracting off the reverse-lag count would not prevent assembly detection on that time scale.

The following figure supplement is available for figure 6:

**Figure supplement 1.** Pipeline of assembly agglomeration algorithm.

distribution is not appropriate [*Gütig et al., 2002*]). From this, and noting that the $M$ binary processes are *not* independent since by construction each higher-rank process $\gamma > \alpha$ can only have '1 s' where the lower-rank processes $\alpha$ had as well (but not necessarily vice versa), one can derive the expectancy value and variance of $\#_{AB,\bar{l}}$, respectively, under the $H_0$ as

$$\mu_{AB,\bar{l}} \equiv E\left[\#_{AB,\bar{l}}\right] = \sum_{\alpha=1}^{M} \frac{\#_A^\alpha \#_B^\alpha}{T - \bar{l}} \qquad (2)$$

$$\sigma^2_{AB,\bar{l}} \equiv E\Big[\big(\#_{AB,\bar{l}} - E\big[\#_{AB,\bar{l}}\big]\big)^2\Big] = \sum_{\alpha=1}^{M} \text{var}\big(\#^{\alpha}_{AB,\bar{l}}\big) + 2\sum_{\alpha=1}^{M-1}\sum_{\gamma>\alpha}^{M} \text{cov}\big(\#^{\alpha}_{AB,\bar{l}}, \#^{\gamma}_{AB,\bar{l}}\big)$$

$$= \sum_{\alpha=1}^{M} \frac{\#^{\alpha}_A \#^{\alpha}_B}{\bar{T}} \frac{(\bar{T}-\#^{\alpha}_A)(\bar{T}-\#^{\alpha}_B)}{\bar{T}(\bar{T}-1)} + 2\sum_{\alpha=1}^{M-1}\sum_{\gamma>\alpha}^{M} \frac{\#^{\gamma}_A \#^{\gamma}_B}{\bar{T}} \frac{(\bar{T}-\#^{\alpha}_A)(\bar{T}-\#^{\alpha}_B)}{\bar{T}(\bar{T}-1)} \text{ with } \tilde{T}=T-\bar{l}.$$

(3)

A parametric test statistic, $S_{\bar{l}} \equiv \big(\#_{AB,\bar{l}} - \hat{\mu}_{AB,\bar{l}}\big)/\hat{\sigma}_{AB,\bar{l}}$ (t-distributed with $2(T-\bar{l})M-1$ degrees of freedom), could be based directly on these moments under the null hypothesis of independence on all time scales (for all choices of $\Delta$), and if the data were truly stationary (by which we mean here that the joint probability distribution $\pi_{AB,t,l}(u,v) \equiv pr\big(c_{A,t}=u \wedge c_{B,t+l}=v\big)$ is time-invariant, i.e. $\pi_{AB,t,l}(u,v) = \pi_{AB,l}(u,v)$ for all $t$). This will commonly not be the case with electrophysiological time series. Rather, there will be rate fluctuations on different temporal scales, as for instance induced by oscillatory drive or external stimuli (*Quiroga-Lombard et al., 2013*). Under these conditions, variance *Equation 3* will usually be highly biased, often underestimating the true variation. Furthermore, the joint count $\#_{AB,\bar{l}}$ may not factor into the product of the marginal counts as in *Equation 2* anymore, since, in general, $E[\#_A]E[\#_{B,l}] \neq (T-l)\sum_{t=1}^{T-l} E[\#_{A,t}]E[\#_{B,t+l}]$ for any *fixed* non-stationary set $\big\{\pi_{AB,t,l}(u,v), t=1\dots T-l\big\}$. Finally, we are interested in testing for *independence at a defined time scale* $\Delta$, but may want to permit the processes to be coupled at other temporal scales, like for instance with *common* external or oscillatory drive. In this case the simple test statistic defined above ($S_{\bar{l}}$) breaks down (*Figure 7* left column). Hence, we would like to test against a stronger $H_0$ which allows for uncoupled or coupled rate changes on broader temporal scales without corrupting our assessment of independence on finer scales.

In the time series literature, the most common remedies for non-stationarity issues are bootstrap-based techniques (*Fujisawa et al., 2008*; *Pipa et al., 2008*; *Picado-Muiño et al., 2013*; *Torre et al., 2013*) and sliding window analyses (*Grün et al., 2002b*).These two methods have, however, severe limitations. Bootstrap-based approaches are computationally quite demanding since essential steps of the algorithm may have to be repeated for a 100–1000 bootstrap replications. This may become outright prohibitive especially when multiple lag constellations are to be considered as in the present work. Sliding window analysis, on the other hand, uses only small fragments of the data set for estimation in each window, thus can be seriously plagued by low sample size issues (resulting in weak statistical power). Sometimes this is (partly) addressed by pooling across many trials, but this in turn requires (a) a sufficient number of trials, (b) stationarity across trials, and (c) clear external time-stamps such that windows across trials are indeed comparable and can be aligned. In many tasks probing higher cognition, where just a handful of trials are not rare (e.g. [*Lapish et al., 2008*; *Hyman et al., 2012*]), or in self-paced tasks, these methods are thus not applicable.

We therefore propose a new approach to non-stationarity here. Rather than testing $\#_{AB,\bar{l}}$ directly for significance, our approach focuses on the asymmetry between the occurrence rates of patterns $(AB,l)$ and $(BA,l) = (AB,-l)$, respectively, defined through

$$\#_{ABBA,\bar{l}} = \#_{AB,\bar{l}} - \#_{AB,-\bar{l}} .$$

(4)

The idea is that non-stationarity on slower time scales ($>\bar{l}\Delta$, if $\Delta$ is the time scale considered), e.g. in the form of correlated rate changes, will, on average, cancel out in $\#_{ABBA,\bar{l}}$, since it will affect $\#_{AB,\bar{l}}$ and $\#_{AB,-\bar{l}}$ alike (for clarity, note that $\#_{AB,\bar{l}}$ is accumulated across $t=1:T-\bar{l}$, while $\#_{AB,-\bar{l}}$ runs across $t=\bar{l}+1:T$; we also note that while differencing to remove non-stationarity is, in general, a more common practice in the 'classical' time series literature, e.g. (*Box et al., 2013*), here we use this technique in a very specific sense by forming the difference between one joint count series and its reverse). More precisely, the scenarios covered by our $H_0(\Delta)$ should be those where the two processes A and B are *independent on the specified scale* $\Delta$, in fact independent for all scales at least up to $l\Delta$, while they may be coupled and/or non-stationary on slower temporal scales of at least $l\Delta$. Our $H_0$ requires $E\big[\#^{\Delta}_{AB,\bar{l}}\big] = E\big[\#^{\Delta}_{AB,-\bar{l}}\big]$ (where superscript $\Delta$ indicates that counts were taken at that temporal resolution), which strictly holds if

$$\sum_{\alpha=1}^{M}\sum_{t=1}^{T-l}p\left(c_{A,t+l}^{\Delta,\alpha}|c_{B,t}^{\Delta,\alpha}\right)p\left(c_{B,t}^{\Delta,\alpha}\right) = \sum_{\alpha=1}^{M}\sum_{\tau=1}^{T-l}p\left(c_{A,\tau}^{\Delta,\alpha}|c_{B,\tau+l}^{\Delta,\alpha}\right)p\left(c_{B,\tau+l}^{\Delta,\alpha}\right) \tag{5}$$

The $H_0(\Delta)$ furthermore demands that this factorizes as

$$\sum_{\alpha=1}^{M}\sum_{t=1}^{T-l}p\left(c_{A,t+l}^{\Delta,\alpha}\right)p\left(c_{B,t}^{\Delta,\alpha}\right) = \sum_{\alpha=1}^{M}\sum_{\tau=1}^{T-l}p\left(c_{A,\tau}^{\Delta,\alpha}\right)p\left(c_{B,\tau+l}^{\Delta,\alpha}\right). \tag{6}$$

Now, if the two processes $A$ and $B$ are reasonably stationary at least on scales up to $l\cdot\Delta$, we have $p\left(c_{A,t}^{\Delta,\alpha}\right)\approx p\left(c_{A,t+l}^{\Delta,\alpha}\right)$ and $p\left(c_{B,t+l}^{\Delta,\alpha}\right)\approx p\left(c_{B,t}^{\Delta,\alpha}\right)$ and the equalities above should (approximately) hold, even if the processes are non-stationary and/or coupled on broader temporal scales. Thus, under the $H_0$ of independence on the time scale fixed by precision $\Delta$, $E\left[\#_{AB,\bar{l}}^{\Delta}\right] = E\left[\#_{AB,-\bar{l}}^{\Delta}\right]$ if (eq. 6) holds exactly, and approximately otherwise.

Under the alternative hypothesis of dependence on the specific scale $\Delta$ considered, on the other hand, if pattern $(AB,\bar{l})$ occurs more frequently than expected by chance, e.g. because unit $A$ excites unit $B$, it appears (mechanistically) rather unlikely that the same is true for the exact time reversed pattern $(AB,-\bar{l})$, unless the two units are in perfect synchrony as treated below (but see sect. 'Choice of reference (correction) lag' where this issue and alternative choices of reference bin are further discussed); hence $\#_{ABBA,\bar{l}}$ would be expected to differ from zero.

To accommodate the strictly synchronous case ($\bar{l}=0$), finally, we slightly modify *Equation 4* to be

$$\#_{ABBA,0} = \#_{AB,0} - \#_{AB,l^*}, \tag{7}$$

with $l^* \neq 0$. While, in principle, different reference lags $l^*$ may be attempted for confirmatory purposes, here we suggest $l^* = -2$ as a tradeoff between potential 'spillover' issues and the effective timescale at which non-stationarity is removed, as discussed in more detail in the sect. 'Choice of reference (correction) lag' below (see also *Figure 1—figure supplement 1*). Note that $\#_{ABBA,0}>0$ if synchrony is the dominant pattern, while the sequential case $\bar{l}=-2$ is tested separately by $\#_{ABBA,-2}$. Again, under the $H_0$, $E\left[\#_{AB,0}\right] = E\left[\#_{AB,l^*}\right]$, while under the $H_1$ of synchronous spiking $\#_{ABBA,0}$ would be expected to be larger than zero (one may also define the symmetric quantity $\#_{ABBA,0} = \#_{AB,0} - \left(\#_{AB,l^*} + \#_{AB,-l^*}\right)/2$, but it makes the computation of the variance, *Equation 10* below, more cumbersome). Following the same lines laid out above in deriving *Equations 2 and 3*, for these modified, stationarity-corrected statistics we thus obtain for the mean $\mu_{ABBA,\bar{l}}$ and variance $\sigma_{ABBA,\bar{l}}^2$ under the $H_0$, respectively,

$$\mu_{ABBA,\bar{l}} \equiv E\left[\#_{AB,\bar{l}}\right] - E\left[\#_{AB,-\bar{l}}\right] = 0 \tag{8}$$

and

$$\sigma_{ABBA,\bar{l}}^2 \equiv E\left[\left(\#_{ABBA,\bar{l}} - E\left[\#_{ABBA,\bar{l}}\right]\right)^2\right] = 2\sigma_{AB,\bar{l}}^2 - 2\mathrm{cov}\left(\#_{AB,\bar{l}},\#_{AB,-\bar{l}}\right), \tag{9}$$

where $\mu_{ABBA,\bar{l}} = 0$ will approximately hold even under non-stationarity (see above). If the process is strongly non-stationary, however, evaluating *Equation 9* directly across the whole time series may still give an inaccurate estimate of variance. In general, we therefore divide the binned spike train into $C$ segments of $k$ time bins each, and combine the local, segment-wise variance estimates into the global estimate $\hat{\sigma}_{ABBA,\bar{l}}^2$ with

$$\hat{\sigma}^2_{AB,\bar{l}} = \mathrm{var}(\#_{AB,\bar{l}}) \quad = \mathrm{var}\left(\sum_{c=1}^{C} \#^c_{AB,\bar{l}}\right) = \sum_{c=1}^{C} \mathrm{var}(\#^c_{AB,\bar{l}}) + 2\sum_{c=1}^{C-1}\sum_{\varsigma>c}^{C} \mathrm{cov}\left(\#^c_{AB,\bar{l}}, \#^{\varsigma}_{AB,\bar{l}}\right),$$

$$\mathrm{cov}(\#_{AB,\bar{l}}, \#_{AB,-\bar{l}}) \quad = \sum_{c=1}^{C} \mathrm{cov}(\#^c_{AB,\bar{l}}, \#^c_{AB,-\bar{l}}) + 2\sum_{c=1}^{C-1}\sum_{\varsigma>c}^{C} \mathrm{cov}(\#^c_{AB,\bar{l}}, \#^{\varsigma}_{AB,-\bar{l}}), \text{ and}$$

$$\mathrm{cov}(\#^c_{AB,\bar{l}}, \#^c_{AB,-\bar{l}}) \quad = \sum_{\alpha=1}^{M} \frac{\#^{c,\alpha}_A \#^{c,\alpha}_B}{k} \frac{(k-\#^{c,\alpha}_A)(k-\#^{c,\alpha}_B)}{k(k-1)^2} +$$

$$+ 2\sum_{\alpha=1}^{M-1}\sum_{\gamma>\alpha}^{M} \frac{\#^{c,\gamma}_A \#^{c,\gamma}_B}{k} \frac{(k-\#^{c,\alpha}_A)(k-\#^{c,\alpha}_B)}{k(k-1)^2}.$$

(10)

For smaller segment length $k$ (here we used $k=100$), this approximation will become more accurate (as *within* each short segment the process will approach stationarity), yet at the same time computationally more demanding. In practice, covariation among segments may often be negligible compared to the within-segment variance contributions (see below; e.g. if auto-correlations decay relatively fast), so to reduce the computational burden one may evaluate the quantity $\hat{\sigma}^2_{ABBA,\bar{l}} \approx 2\sum_{c=1}^{C} \mathrm{var}(\#^c_{AB,\bar{l}}) - 2\sum_{c=1}^{C} \mathrm{cov}(\#^c_{AB,\bar{l}}, \#^c_{AB,-\bar{l}})$. This is in fact the estimate we have used throughout this manuscript.

Based on the estimates derived above, we can then define the following approximately *F*-distributed quantity which can be used for significance testing:

$$Q_{\bar{l}} \equiv \frac{\left(\#_{ABBA,\bar{l}} - \mu_{ABBA,\bar{l}}\right)^2}{\hat{\sigma}^2_{ABBA,\bar{l}}} \sim \mathrm{F}_{1,\nu} \tag{11}$$

with 1 numerator and $\nu$ denominator degrees of freedom. We found these approximations to work reasonably well for $E\left[\#_{AB,\bar{l}}\right]>4$ (see *Figure 7*). For one numerator and large denominator d.f., the *F* distribution is known to converge to the $\chi^2_1$ distribution which could be used for testing instead. For smaller sample sizes and non-stationary scenarios (where the variance $\hat{\sigma}^2_{ABBA,\bar{l}}$ is not exact anymore but estimated from segments), however, the *F* distribution appeared more appropriate, although the exact denominator d.f. are unknown in this case. For all analysis reported here we have used $\nu = 2(T-\bar{l})M - 1$, but more generally we recommend $\nu = T - \bar{l}$ which is more conservative for low sample size ($T<50$; the differences become negligible for $T>400$ as in the case of most analyses reported here). Finally, all $\alpha$-levels are Bonferroni-corrected for the number $R$ of tests performed (see pseudo-code below and sect. '*Recursive assembly agglomeration algorithm*').

## Limitations of parametric testing under non-stationarity

To examine the error made by the various approximations introduced above, we empirically studied different scenarios by simulation. In one set of simulations, discrete, step-like rate-changes were used. Within a total of $T=10^6$ 'elementary' time bins (not to be confused with bin width $\Delta$ used for assembly detection), low-rate states were randomly interspersed with $m$ high-rate states of duration $L$ (expressed in terms of numbers of elementary bins). For each elementary time bin, spikes were drawn from a Bernoulli process with probabilities $\pi_{\mathrm{low}}$ and $\pi_{\mathrm{high}}$, respectively. Here, $L$ explicitly defines the time scale on which the two processes are non-stationary and, in some simulations, coupled. Simulations (with $\Delta=100$, $l\in\{0,5,10\}$) were performed with both relatively fast ($m=25$, $L=3000$, on the order of $2l_{max}\Delta$; *Figure 7—figure supplement 1*, center column) and slow ($m=1$, $L=75000$; *Figure 7—figure supplement 1*, right column) rate variations (both with the same number of high-rate bins), and with one independent scenario (*Figure 7—figure supplement 1*, top row) and one where the rate variations were completely synchronized between the two processes (*Figure 7—figure supplement 1*, bottom row). As the percentile-percentile plots in *Figure 7—figure supplement 1* reveal, in all these cases departures from the theoretical *F* distribution were relatively mild. We emphasize that this was the case although the difference between the low and high rate states was assumed to be rather large in our simulations ($\pi_{A,\mathrm{low}}=0.01$ vs. $\pi_{A,\mathrm{high}}=0.05$, and $\pi_{B,\mathrm{low}}=0.03$ vs. $\pi_{B,\mathrm{high}}=0.15$), a larger violation of stationarity than one may actually expect empirically.

In another set of simulations, time-varying firing rates for the two neurons were drawn from a slowly varying auto-regressive process with Gaussian noise (or, equivalently, a joint multivariate Gaussian). Spike counts for each bin were then drawn from a Poisson distribution with the rate $\lambda$

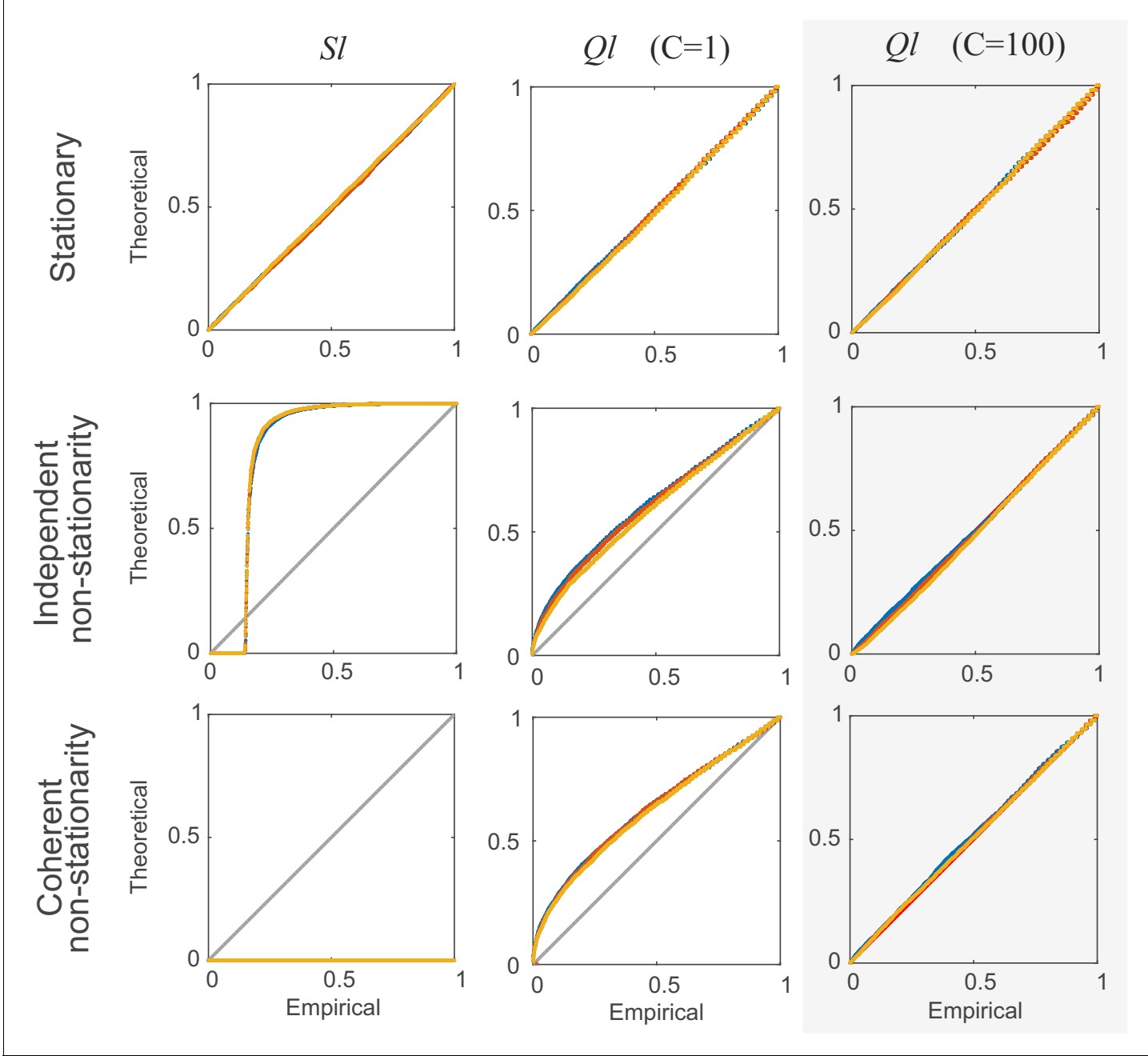

**Figure 7.** Comparison of non-corrected ($S_i$) and stationarity-corrected ($Q_i$) pairwise test statistics. Percentile-percentile plots showing agreement between the theoretical distributions for different test statistics considered in the text ($S_i$, $Q_i$ with $C$=1, and with $C$=100 segments) and the distributions empirically obtained, for the truly stationary case (top row), independent non-stationarity (center row), and non-stationarity coupled among the two units $A$ and $B$ (bottom row). Overlaid are distributions derived from 4000 simulation runs with spike time series analyzed for the three different lags $l$= 0 (blue curves), 5 (yellow) and 10 (red). $\Delta$=100 in all cases. Simulations are with non-stationarity implemented as step-type rate-changes (see *Materials and methods*) with $m$=1 and $L$=75000. Identity line (bisectrix) is marked in gray. Results for test statistic used for all data analyses in this work highlighted by light-gray box.

The following figure supplements are available for figure 7:

**Figure supplement 1.** Statistical testing under non-stationarity on different time scales: step-like rate change.

**Figure supplement 2.** Statistical testing under non-stationarity on different time scales: rate covariation.

*Figure 7 continued on next page*

*Figure 7 continued*

**Figure supplement 3.** Detecting coupling among oscillating units.

determined by the auto-regressive process passed through a non-negative transform ('link-function', see *Equation 13* below). We simulated scenarios with both somewhat faster ($cov(\lambda_{A,t}, \lambda_{A,t+2\Delta l_{max}}) \approx 0.8$; *Figure 7—figure supplement 2*, center column) and very slowly decaying auto-correlations ($cov(\lambda_{A,t}, \lambda_{A,t+2\Delta l_{max}}) \approx 0.99$; *Figure 7—figure supplement 2*, right column), and without ($cov(\lambda_{A,t}, \lambda_{B,t}) = 0$; *Figure 7—figure supplement 2*, top row) or with ($cov(\lambda_{A,t}, \lambda_{B,t}) \approx 0.7$; *Figure 7—figure supplement 2*, bottom row) correlations among the two processes present. As *Figure 7—figure supplement 2* reveals, the results for these simulation experiments were very similar to those shown in *Figure 7—figure supplement 1*, with the empirical $Q_{\bar{l}}$ distribution deviating only slightly from the theoretical $F$ distribution. We thus conclude that for empirically reasonable scenarios of non-stationarity and coupling on longer time scales, the parametric statistical procedure introduced above should return quite accurate results.

It is important to note that while coherent rate changes constitute a coupled non-stationarity from the viewpoint of smaller timescales $\Delta$, for which they will be removed by our difference statistic, they actually represent an assembly on their own characteristic scale (type V in *Figure 1*). They are indeed correctly identified as such in both simulated scenarios when the bin width is chosen to be about the same as the time-scale of the 'non-stationarity', i.e. $\Delta \approx L$ ($m$=75, $L$=1000; *Figure 7—figure supplements 1,2*, left columns, blue curves in bottom graphs). Hence, non-stationarity, in the present definition, refers to somewhat slower (co-)variations that may mislead detection of coincident events on comparatively finer time scales. Both, the relevant time scale for assembly detection and the associated time scale of non-stationarity, are strictly defined by the experimenter by fixing $\Delta$. Whether the detected temporal structure is interpreted as an assembly or as non-stationarity therefore depends on the timescale chosen, and is ultimately up to the experimenter and the research question posed. From this perspective, our method basically ensures that coincidence structure is not falsely attributed to a certain temporal precision $\Delta$ while it was really produced by slower (co-) variations, an issue that has frequently plagued the discussion about precise temporal coding in the nervous system in the past (eg., [*Singer, 1999*] vs. [*Shadlen and Movshon, 1999*]).

As another note of caution, we remark that while under the $H_0$ assumptions (independence and stationarity on scales $\leq l\Delta$) equality (*Goldman-Rakic, 1995*) will always hold (in expectancy), the size of the deviations from the $H_0$ in the case of true structure (and thus the test's sensitivity or power) may, in principle, depend on how exactly non-stationary processes interact with underlying structure-creating processes (e.g., linearly vs. non-linearly).

## Oscillations

Finally, as an example of a particularly common form of non-stationarity in neural data (e.g. [*Quiroga-Lombard et al., 2013*]), we considered oscillations (*Buzsáki and Draguhn, 2004*). (Note that oscillations constitute a type of non-stationarity from the present neurophysiological perspective, although they may be considered stationary in the classical statistical definition with access to an infinite ensemble of time series starting at random phases (see [*Fan and Yao, 2003*]). We tested two scenarios here: One in which two neurons were spiking independently at the time scale considered, but were driven by a common oscillatory drive at the same frequency and phase, and one where on top the units exhibited supra-chance coincident patterns. Specifically, for both units $A$ and $B$ the firing probability was taken to follow a Poisson distribution with rate parameter $\lambda_{\{A,B\}}(t) = 5(\sin(2\pi\theta t)0.6 + a_{\{A,B\}})$, $a_A = 1$, $a_B = 0.5$, $\theta = 4Hz$, yielding mean firing rates of 5Hz and 2.5Hz, respectively. For the independent case, no structure is detected for smaller bin widths $\Delta$ up to $\approx$30 ms (*Figure 7—figure supplement 3A*), while for larger bin widths (>50 ms) the algorithm picks up the coordinated rate changes. For the second scenario (finer time-scale dependence on top of oscillation), patterns are inserted on top with a spike in unit $A$ followed by one in unit $B$ 20 ms later, phase-locked to the underlying rhythm. Within a $T$=1500 s long simulation period, a total of 90 of such patterns were placed randomly with a 20 ms delay to the oscillation peak. *Figure 7—figure supplement 2B* shows that these patterns are indeed correctly detected by our procedure at the

smaller bin sizes ({5, 10} ms) tested. Thus, once again, this example illustrates that the detection of finer time scale spike time relations is not confounded by covariations at slower scales, while the slower covariations are flagged up as a coherent structure at their own respective time scale (50–120 ms).

## Choice of reference (correction) lag

To eliminate non-stationarity as a confounding factor, we suggested computing the difference between the target-lag joint count $\#_{AB,l}$ and its time inverse $\#_{AB,-l}$. Strictly, this implies that if there were precise repeats of spiking sequences in reverse order, both the forward and backward directions would go undetected as they would cancel each other. Although a few studies have suggested that reverse replay may indeed occur in hippocampus (*Foster and Wilson, 2006*; *Diba and Buzsáki, 2007*), in general it seems unlikely that sequential patterns are replayed in reverse order with *exactly* the same time lag constellation and at the very *same temporal scale* as in the forward direction, such that the patterns would fully cancel (or at least this seems not more likely than constellations implied by any other choice of reference lags). Nevertheless, this brings up the more general question of whether there is a better choice of reference lag. Furthermore, the pairing of $l$ with its exact opposite, $-l$, implies that non-stationarity will be removed at different timescales (with different levels of precision, $l\Delta$) for different lags, a potentially undesired effect if direct comparisons between structures with different lag sizes are sought.

Indeed, in principle, any other lag might potentially be chosen as a reference. A general recommendation therefore might be to simply repeat the analyses with other reference lags if researchers suspect that there might be significant structure at both $\bar{l}$ and $-\bar{l}$, or to fix the reference lag to be, e. g., $l^* = \pm(l_{max} + 1)$, with $l_{max}$ the maximum lag tested for, for *all* lags considered. Doing this for the ACC delayed alternation data set presented in *Figure 4* as an example, we found that the overlap between assemblies detected with $l^* = -\bar{l}$ and $l^* = \pm(l_{max} + 1)$ (fixed for all query lags) was on average $\approx 97\%$ across all timescales (range 96–99%; as measured by the Rand index, *Equation 14*, defined across all matching pairs with significant $[r]$ or non-significant $[s]$ relation), including the synchronous ($\bar{l} = 0$) case.

This does not imply that the reference lag is arbitrary, however. In general, there are two factors to consider: The amount of non-stationarity permitted (loosely related to the type I error in statistical terms) vs. the true structure potentially removed by the choice of reference lag (related to the test's sensitivity or 'type II error'). For instance, while choosing a directly neighboring bin as reference, $l^* = \bar{l} - 1$, may, at first glance, appear like the ideal choice from the perspective of removing non-stationarity confounds most efficiently (since in this case we only require $p\left(c_{B,t}^{\Delta,\alpha}\right) \approx p\left(c_{B,t+1}^{\Delta,\alpha}\right)$ for *Equation 6* to hold), it may also imply the highest risk of removing true structure from both a physiological and statistical point of view (the more similar and temporally proximal target and reference statistics, the more likely they are to be correlated). This is illustrated for the important case of synchronous activity on simulated data (type I & V assemblies) in *Figure 1—figure supplement 1*, where with $l^* = \bar{l} - 1$ we find that the test starts to lose part of the structure due to spillover into neighboring bins, while for too large lags the test's temporal accuracy goes down, and with that, more generally, the false discovery rates due to non-stationarity would be expected to rise. Furthermore, given that by far most synaptic connections and physiological responses are unidirectional or at least asymmetrical (*Markram et al., 1997*), the scenario that a neuron *A* is correlated with a neuron *B* at a *couple of different forward lags* (e.g., due to bursting) appears more likely than a pair of asynchronous neurons strictly switching order multiple times. Finally, the strict symmetry implied by $l^* = -\bar{l}$ makes the joint count statistics for target vs. reference lag strictly comparable (unspoiled, e. g., by possibly differing auto-correlations across the target vs. reference lag), and permits to interpret *Equation 11* effectively as a two-sided test (with directionality indicated by the sign of $\#_{ABBA,\bar{l}}$).

While the ideal choice of reference lag may be an issue of further theoretical and empirical investigation, we emphasize that a), in practice, the precise choice of reference lag should not be overly crucial (as supported by the analyses reported above and in *Figure 1—figure supplement 1*), and b) analyses may always be repeated for a few different reference lags if in doubt about structure possibly missed by the initial choice of reference.

## Recursive assembly agglomeration algorithm

Our assembly agglomeration scheme starts from all significant pair-wise interactions, and then adds new elements only on the basis of the structures already formed, similar in spirit to the apriori-algorithm in machine learning (*Hastie et al., 2009*; *Sastry and Unnikrishnan, 2010*). This heuristic procedure drastically reduces the number of configurations to be tested, but may lose significant unit configurations with non-significant subgroups (*Picado-Muiño et al., 2013*). For each pair of units $A$ and $B$, the spike count $\#_{AB,l}$ is obtained for each triplet $(AB, l)$, with $l \in \{-l_{max} \ldots l_{max}\}$, and the maximum count is tested for significance (*Figure 6—figure supplement 1*, step 2). Since the marginal elementary processes $\#_A^\alpha$ and $\#_B^\alpha$ do not change for different lags $l$ (except for the small influence from the $l$ bins cut off), this selection procedure is formally equivalent to performing an explicit significance test for each lag $l$ and retaining the one associated with the lowest $p$ value. In the next step, all significant configurations $(AB, \bar{l})$ are treated like single units, with the joint 'spike (activation) times' defined (arbitrarily) as those of unit $A$ whenever it matches up with a spike in unit $B$ separated by $\bar{l}$ time steps (bins). Each significant pair $(AB)$ is then paired in turn with all single units $C$ which had entertained a significant relationship with either $A$ or $B$ in the previous step (*Figure 6—figure supplement 1*, step 3). Proceeding with composite pairs $\left((AB, \bar{l}_{AB})C, \bar{l}_{(AB)C}\right)$ exactly as described above, higher-order structures are thus recursively built up. Note that this procedure effectively tests for higher order structure, rather than just aggregating pairwise information: After screening for pairwise relations in the first step, for instance, in the second iteration the algorithm tests for the factorization $P(A, B, C) = P(A, B)P(C)$ whenever a unit $C$ is considered for inclusion into the already formed set $(AB, \bar{l})$, rather than, e.g., $P(A, B, C) = P(A)P(B)P(C)$, or "$P(A, B) = P(A)P(B) \vee P(A, C) = P(A)P(C) \vee P(B, C) = P(B)P(C)$". Likewise, higher order joint distributions are considered in all subsequent iterations.

Significance levels $\alpha$ at each step of the agglomeration scheme are strictly Bonferroni-corrected as $\bar{\alpha}_i = \alpha/R_i$ (using $\alpha = 0.05$ here), with $R_i$ the total number of tests performed. Specifically, for the first step, $R_1 = N(N-1)(2l_{max}+1)/2$, where $N$ is the total number of single units (correcting for the total number of different pairs). For each subsequent step $i$, $R_{i,a} = N_{a,i}N_{u,a}(2l_{max}+1)$, where $N_{a,i}$ is the number of assemblies tested in iteration $i$, and $N_{u,a}$ the number of units tested in combination with that assembly $a$ (hence allowing an higher $\alpha$-level for assemblies tested in conjunction with less different units). At any step, unit-sets with the same elementary units but different lag distributions may result. From all these, we select only the one associated with the lowest $p$-value, and discard all others. This whole procedure will stop when no units engage in significant relationships anymore with the already agglomerated sets (*Figure 6—figure supplement 1*, step 4). All true subsets of larger sets are finally discarded (but may be retained if hierarchical nesting is of interest, see below). A pseudo-code for the agglomeration scheme is included below. To test for structure at different temporal resolutions (scales), the whole scheme is re-run for a range of user-provided bin widths $\Delta = \{\Delta_{min} \ldots \Delta_{max}\}$. For each assembly pattern the width $\Delta^*$ associated with the lowest $p$-value is defined as the characteristic time scale (or temporal precision) for that assembly.

As a final note, for very large $\Delta$, the binned spike counts may potentially fluctuate around a high mean level and never fall below some minimum count $c_{floor}$ considerably larger than zero for the whole time series. In our count statistics, also spikes up to that baseline rate $c_{floor}$ would contribute to the coincidence counts $\#_{AB\bar{l}}$, although they are completely non-informative with respect to the coupled dynamics among units, thus potentially biasing the results for large $\Delta$. In this case, since we are only interested in actual firing rate covariations, we suggest to subtract off the minima $min\left(\{c_A^\Delta\}_t\right)$ and $min\left(\{c_B^\Delta\}_t\right)$ from the two considered series $A$ and $B$, respectively, thus removing the non-informative floor count before statistical testing. This procedure would not affect the evaluation of spike coincidences at reasonably small $\Delta s$ for which $min\left(\{c^\Delta\}_t\right) = 0$, obviously.

## Further assembly pruning

Further pruning may be applied to the set of assemblies returned by the algorithm if desired. This may sometimes help interpretability and visualization, but of course depends on the exact analysis goals. If, for instance, the interest is in whether the same assemblies are replayed at a different time scale (e.g. [*Diba and Buzsáki, 2007*]), then one may want to keep more than just the one assembly associated with the lowest $p$-value across time scales $\Delta$. Here, solely for the purpose of visualization,

in *Figure 3A* the full set of assemblies returned by the algorithm was pruned by selecting among all assemblies (across different Δ) with cosine distance <0.3 only the one with lowest *p*-value. In *Figure 1B*, again for clarity and visualization, pruning was performed by discarding across scales Δ any assembly which is a subset of another, larger assembly (by default this is always done *within* each time scale Δ). No pruning was used for any of the other figures presented in here.

## Assembly activation

An instance of assembly activation in the multivariate spike time series was registered whenever spikes in the elementary assembly units occurred in the order prescribed by the associated pattern of time lags, with the activation time point defined as that of the assembly unit spiking earliest. The total assembly activation score (as given in *Figure 2B–C*) is then defined as the number of such activation instances within a given time bin of size Δ. This can lead to activation scores much larger than one, especially for assemblies defined through rate changes on coarser time scales, since each set of single assembly unit spikes occurring in the right order is counted. (Alternatively, one may define assembly activation through the correlation of the average assembly spike count pattern with the observed spike count patterns along the series of binned spike counts at the respective assembly resolution Δ. This would result, however, in a temporally much less well resolved activation score which otherwise would essentially return the same information.)

A Matlab (MathWorks) implementation of the whole procedure is provided at https://www.zi-mannheim.de/en/research/departments-research-groups-institutes/theor-neuroscience-e/information-computational-neuroscience-e.html. To give an idea of the performance speed, on a 12-core, 2.5 GHz, workstation, for a set of 50 simulated units (see below), a time series of length $T$=1400 s, and with $\Delta = \{0.015, 0.05, 0.1, 0.15, 1\}$s and $l = \{-10, \ldots, 10\}$, this whole procedure took <50 min for five embedded assemblies with five units each (scenario from *Figure 1*). Significant further improvements in performance speed may be obtained through an implementation in a more basic language like C++.

## Pseudo-code for agglomerative assembly formation

```
% N: total number of units
% u_i, i=1...N: single units
% U_m: set of units and corresponding lags (assemblies)
% r: set counter

for i = 1:N, U_i ← {(u_i, 0)}                    % Initialize lists with single units u_i
for all i ≤ N, j ≤ N: Z_ij = FALSE               % Initialize all single unit pair comparisons to be
                                                 'false' (= 'accept H_0')
r = N, L^old = 0

REPEAT                                           % agglomeration procedure

L^new = r
for  m = L^old + 1:L^new                         % move through all lists formed in previous step
      for all u_s ∉ U_m | m < s ≤ N ∨ (m > N ∧ ∃ u_l ∈ U_m: Z_sl = TRUE)
      % in first step (m ≤ N) probe U_m with all other single units not yet tested, or (for
      % m>N)  probe  U_m  with  all  other  single  units  that  occur  in  at  least  one  other
      % significant pair with a unit from U_m

      l̄ ≡ argmax_l (#_(u_m,us),l) % test for significance at lag l̄ with maximum pair-wise count:

      if Pr(Q_l̄ ≥ F_1,(T−l̄)M−|_1[U_m, u_s, l̄]|H_0) ≤ α / R with R = (L^new − L^old) · |{u_s}| · (2l_max + 1):
          r ← r + 1,
          U_r ← {U_m, (u_s, l̄)} = {(u_r,1, 0), (u_r,2, l̄_2), ..., (u_r,|U_m|+1, l̄_|U_m|+1)}
          if |U_m| = 1, Z_sm = Z_ms = TRUE
          % form new list where each l̄_j is defined relative to the activationtime point ('0') of
          the first unit u_r,1 in the ordered list; set pair-wise flag to 'true' if single-unit
```

```
          comparison
L^old ← L^new
UNTIL Pr(Q_l̄ ≥ F[U_m, u_s, l̄]|H_0) > α /R for all m, s

% Pruning steps:
Discard all U_m for which ∃ n ≠ m: U_m ⊂ U_n For all U_n, U_m for which ∀ u_s ∈ U_n: u_s ∈ U_m :Remove U_m if Pr
(Q_l̄ ≥ F[U_m]|H_0) > Pr(Q_l̄ ≥ F[U_n]|H_0), and  U_n otherwise
```

In the algorithm above |·| denotes the cardinality of a set, and all set-operations ($\in$, $\subset$, etc.) are defined in terms only of the unit-elements composing a set (i.e., ignoring the associated lags with which their occur).

## Alternative procedure

In the REPEAT-loop, instead of probing all pair-wise relations among the current lists (assemblies) and all *single units* from significant pairs, one could also check for significant relationships among *pairs of lists* $U_n, U_m$. As in classical hierarchical, agglomerative cluster-analytic procedures (*Gordon, 1999*), at each step one may only fuse the pair ($U_n, U_m$) associated with the lowest *p*-value, add this to the current set of lists while removing $U_n, U_m$, and repeat until $Pr(Q_{\bar{l}} \geq F[U_n, U_m, \bar{l}]|H_0) > \alpha/R$ for all *n*, *m*. This would yield a dendrogram-like representation and thus reveal strictly *hierarchical nesting* among the assemblies. It comes at the cost, however, that a) many higher-order assemblies may go undetected, and b) partial overlap among assemblies, which one may expect if the units in assemblies act like 'letters in words', would be prohibited by the definition of the agglomerative procedure. Also note that hierarchical nesting, to the degree present, could also be revealed with the definition of the agglomeration scheme in the pseudo-code above if subsets of further agglomerated sets are not pruned away at the end.

## Construction of synthetical 'ground-truth' data

To test the full assembly detection schemes developed above, artificial spike trains from 50 cells were created according to inhomogeneous Poisson processes by drawing inter-spike-intervals from an exponential distribution with rate parameter $\lambda_{it}$ for each unit *i*. Instantaneous firing rates $\lambda_{it}$ were governed by an underlying stable first-order autoregressive process

$$\mathbf{s}_{t+1} = \mathbf{D}\mathbf{s}_t + \boldsymbol{\varepsilon}_t, \quad \boldsymbol{\varepsilon}_t \sim N(\mathbf{0}, \sigma_s^2 \mathbf{I}) \tag{12}$$

with coefficient matrix $\mathbf{D}$, and $E\left[\boldsymbol{\varepsilon}_t \boldsymbol{\varepsilon}_{t'}^T\right] = 0$ for all $t \neq t'$ (white noise process). (Note that although $\mathbf{D}$ is set such that the autoregressive process itself is stationary, i.e. $max(|eig(\mathbf{D})|) < 1$, it implies fluctuations in the firing rate which makes the Poisson spiking processes themselves non-stationary in our definition above). Since $\mathbf{s}_t$, in principle, is unbounded (in particular, can assume negative values), it was pushed through a sigmoid non-linearity

$$\lambda_t = \left(1 + erf\left(\upsilon \frac{\mathbf{s}_t - \bar{\mathbf{s}}}{\sigma_s}\right)\right)\bar{\lambda}, \tag{13}$$

with *erf* the error function, and constant mean rate vector $\bar{\lambda}$. Finally, to ensure a refractory period, a constant delay $\tau_{ref}$ is added to each inter-spike-interval. Where not indicated otherwise, parameters used for the simulations were $\bar{\lambda} = 5Hz$ for all units, $\tau_{ref} = 15ms$, $\mathbf{D} = 0.9\,\mathbf{I}$, where $\mathbf{I}$ denotes the identity matrix, $\upsilon = 0.2$, $\sigma_s = 0.01$.

Assemblies of all five types illustrated in *Figure 1A* were embedded within the same set of 50 spike trains as disjunctive groups of 5 neurons each. Note that since our algorithm is aimed at detecting significant spike time patterns (rather than, for instance, underlying connectivity), explicit control of such patterns and spike train statistics with vivo-like characteristics is most important for a ground truth check, while adding more biophysical realism to the underlying simulation setup would not help in this case. For assembly type I, each occurrence is marked by five precisely synchronous spikes across the set of assembly neurons (e.g. [*Harris et al., 2003*; *Miller et al., 2014*]). For assembly type II, spikes follow a precise sequential pattern across the set of assembly neurons on each instance of activation (*Lee and Wilson, 2002*; *Diba and Buzsáki, 2007*). Time lags between spikes

were drawn from a uniform distribution [0 0.1] s, and then fixed for each occurrence. For assembly type III, spikes across the set of assembly neurons followed a precise temporal pattern, but did not exhibit a strict temporal order, i.e. each neuron could contribute one to several spikes to the assembly pattern without strictly leading or following others (e.g. [*Ikegaya et al., 2004*]). For the simulations, these patterns were generated by distributing a few spikes at a Poisson rate of 10 Hz across a period of 0.2 s for each assembly neuron, but then keeping these patterns fixed on each occasion of assembly activation.

For the less precise assembly type IV, short windows of extra spikes for each assembly neuron were organized in a specific temporal pattern, with the exact occurrence of the extra spikes within the defined time windows determined randomly on each repetition (cf. *Figure 1A*; e.g. [*Friedrich et al., 2004*; *Euston et al., 2007*; *Luczak et al., 2007*; *Peyrache et al., 2009*; *Adler et al., 2012*]). Specifically, time windows of 0.3 s with extra spikes at a Poisson rate of 10 Hz were (without loss of generality) arranged in a sequential order, with the time lag between these windows drawn from a uniform distribution, [0 0.4] s. While this sequential ordering of time windows was fixed, within each window spikes were drawn at random on each assembly repetition. Assembly type V, finally, was simply defined by an increase of the Poisson firing rate from 5 Hz to 10 Hz for periods of 1 s simultaneously within the set of assembly neurons, as, e.g., during the delay period of a working memory task (e.g. [*Fuster, 1973*]).

For all assembly patterns, all spikes from the background process were erased within a $\pm 15 \, ms$ window around each assembly spike to preserve the refractory period. Assembly activation times were distributed (uniformly) randomly across the whole spike time series.

## Performance evaluation: Low sample size limit and corrupted spike trains

To evaluate the performance, statistical power, and potential biases of our assembly detection algorithm more systematically, we focused on two experimentally relevant scenarios: Low assembly occurrence rates and spike sorting errors. The Rand index (*Rand, 1971*) was used to quantify the match between predefined assemblies and those retrieved by the algorithm. The Rand index measures the agreement between two partitions, in our case of units into assemblies, and is defined as

$$R(r,s) = \frac{r+s}{n(n-1)/2} \tag{14}$$

where $r$ is the number of unit pairs correctly assigned to the same assembly in both partitions, $s$ the number of unit pairs correctly assigned to two different assemblies, and $n$ the total number of detected assembly units. $R(r,s)$ varies between 0 and 1, assuming 1 only if the assembly structure extracted by the algorithm exactly maps onto the one predefined. To obtain a clean picture on the algorithm's statistical performance for each assembly type, unconfounded by the presence of other assembly types, in these analyses each assembly type from *Figure 1* was investigated separately (i. e., unlike the analyses described in the main text where the different assembly types were mixed in the same simulations).

*Figure 2A* plots $R(r,s)$ for all five assembly types from *Figure 1* as a function of total assembly occurrences (in spike time series of length $T$=1400 s). Obviously, assembly detection gradually degrades as these structures start to drown in the noise but, as *Figure 2A* reveals, this only happens when the occurrence rates drop below ~0.18 repetitions/sec. Likewise, as shown in *Figure 2B*, more than ~30% of all spike times need to be corrupted by spike sorting errors (assignment of assembly spikes to wrong units) before performance notably decays. In more detail, *Figure 2A and B* suggest that sequential assembly patterns (types II and IV) are more vulnerable to lower sample sizes than those assemblies defined through precisely or broadly, respectively, synchronous firing (types I and V). The likely reason for this is that the binning procedure itself introduces some noise (as spikes may fall randomly into one or the other of two neighboring bins), which affects sequential assemblies more than simultaneous assemblies (for which, in the case of our simulations, it is guaranteed that aligned groups of spikes end up in the same bin). Considering just assembly types I and V, also note that assemblies defined by broader simultaneous rate increases are detected much more easily than those characterized by precise spiking. This is not surprising given that in absolute terms each incidence of an assembly of type V contributes much more spikes to the whole process than an

assembly of type I, thus in turn causing assemblies of type V being detected across a much larger range of bin widths than assemblies of type I (cf. *Figure 1C*). Perhaps most importantly, however, as studied in *Figure 2C–D*, our statistical framework is quite conservative and rarely produces false positives in the simulated scenarios, with a false discovery rate (fraction of units incorrectly assigned to an assembly) mostly remaining below or around 0.5% across all conditions examined.

## Experimental procedures

The in-vivo recordings from the rat (Long-Evans) anterior cingulate cortex (ACC) were taken from two studies by Hyman et al. (*Hyman et al., 2012*; *Hyman et al., 2013*). In both studies, multiple single unit recordings were performed with a set of 16 simultaneously implanted tetrodes, with an average of 35 and 30 isolated (and artifact-free) units per recording session for the environmental exploration and delayed alternation task, respectively (with $n=9$ and $n=11$ sessions in total). In the environmental exploration task studied in Hyman et al. (*Hyman et al., 2012*), rats were offloaded in a novel environment which they were free to explore, with one to several transfers between two different environments. Each environment was analyzed separately by concatenating the spike trains associated with the repeated exposures to the same environment. The delayed alternation task studied in Hyman et al. (*Hyman et al., 2013*), a classical working memory paradigm, took place in a Skinner-box with two levers which the animals had to press in alternating fashion. A delay of 10 s was introduced between each lever press and a nose poke the animals had to perform on the side opposite to the levers before continuing with the next lever press.

Hippocampal and entorhinal cortex (EC) recordings on the exploration task, performed simultaneously within these two areas, were borrowed from (*Mizuseki et al., 2013*). Recordings were collected from three Long-Evans rats implanted with multi-shank (32 or 64 sites) silicon probes lowered into the CA1 hippocampal pyramidal layer and into layers 3–5 of entorhinal cortex. In this task (*Mizuseki et al., 2009*), rats were free to explore a 180 cm x 180 cm arena with water or Froot Loop items randomly dispersed throughout. Here we analyzed $n=28$ sessions (selecting always the longest session from each day) with on average 22 (CA1) and 19 (EC) artefact-free units per session, respectively. CA1 and EC recordings on the delayed alternation task come from (*Pastalkova et al., 2008*), who used the same animals employed on the exploration task (*Mizuseki et al., 2009*), from which we took $n=23$ sessions (again selecting the longest from each day) with on average 28 (CA1) and 22 (EC) isolated and reasonably artefact-free units. Animals had to alternate between the two arms of a figure-eight shaped maze to obtain reward at water spouts located at the rear of the arms. A delay of 10 or 20 s, respectively, spent in a running wheel, was inserted between trials for the two animals tested. For all analyses, all units with average firing rates below 0.2 Hz were excluded. Please see original publications for further details on electrode placement, unit separation, and experimental design. CA1 and EC datasets are publicly available at www.crcns.org; ACC datasets will be made available at https://www.zi-mannheim.de/en/research/departments-research-groups-institutes/theor-neuroscience-e/information-computational-neuroscience-e.html.

## Acknowledgements

We are deeply indebted to Drs. James Hyman and Jeremy Seamans for lending us their ACC data (reported in [*Hyman et al., 2012*; *Hyman et al., 2013*]) for the present analysis. The CA1 and EC data were made publicly available by the authors (*Mizuseki et al., 2013*) at www.crcns.org. We also would like to thank Drs. Andreas Draguhn, Martin Both, Thomas Fucke, Hazem Toutounji, Loreen Hertäg, and Grant Sutcliffe for commenting on a previous version of this article.

## Additional information

### Funding

| Funder | Grant reference number | Author |
| --- | --- | --- |
| Deutsche Forschungsgemeinschaft | CRC-1134, SP D01 | Daniel Durstewitz |
| Bundesministerium für Bildung und Forschung | 01GQ1003B | Daniel Durstewitz |

| Deutsche Forschungsge-meinschaft | DU-354/8-1 | Daniel Durstewitz |

The funders had no role in study design, data collection and interpretation, or the decision to submit the work for publication.

### Author contributions

ER, DD, Conception and design, Analysis and interpretation of data, Drafting or revising the article

### Author ORCIDs

Eleonora Russo, http://orcid.org/0000-0002-6215-6305
Daniel Durstewitz, http://orcid.org/0000-0002-9340-3786

## Additional files

### Major datasets

The following previously published dataset was used:

| Author(s) | Year | Dataset title | Dataset URL | Database, license, and accessibility information |
| --- | --- | --- | --- | --- |
| Mizuseki K, Sirota A, Pastalkova E, Diba K, Buzsáki G | 2013 | Data from: Multiple single unit recordings from different rat hippocampal and entorhinal regions while the animals were performing multiple behavioral tasks | http://dx.doi.org/10.6080/K09G5JRZ | Available at Collaborative Research in Computational Neuroscience (http://crcns.org/) |

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

**Appendix**

## Relation to previous methodological approaches

Numerous other statistical procedures for detecting assemblies or sequential patterns have been previously proposed (*Abeles and Gerstein, 1988*; *Abeles and Gat, 2001*, *2001b*; *Tetko and Villa, 2001a*; *Grün et al., 2002a, 2002b*; *Harris, 2005*; *Pipa et al., 2008*; *Sastry and Unnikrishnan, 2010*; *Staude et al., 2010a, 2010b*; *Humphries, 2011*; *Lopes-dos-Santos et al., 2011*; *Gansel and Singer, 2012*; *Gerstein et al., 2012*; *Shimazaki et al., 2012*; *Picado-Muiño et al., 2013*; *Torre et al., 2013, 2016a*; *Lopes-dos-Santos et al., 2013*; *Billeh et al., 2014*; *Logiaco et al., 2016*), but most of these adhere to one or the other theoretical conceptualization of a cell assembly (cf. *Figure 1A*), or become (computationally) impractical for larger cell numbers or multiple lags, e.g. because they rely on time-consuming bootstrap analyses (e.g. [*Abeles and Gat, 2001*; *Pipa et al., 2008*; *Fujisawa et al., 2008*; *Gansel and Singer, 2012*; *Picado-Muiño et al., 2013*; *Torre et al., 2013, 2016a*]).

Along similar lines as our procedure, unitary event analysis scans simultaneously recorded spike trains for precise spike co-occurrences (*Figure 1A,I*) that exceed the joint spike probability predicted from independent Poisson processes with the same local rate (*Grün et al., 2002a, 2002b*). However, this procedure has not been extended yet to multiple lags (but see *Torre et al. (2016a)*) or larger bins (with higher counts), and deals with non-stationarity through sliding windows or bootstrap analyses. In another approach to synchronous spike-cluster detection based on the cumulants of the population spike density of all simultaneously recorded neurons, Staude et al. (*Staude et al., 2010a, 2010b*) developed a method and stringent statistical test for checking the presence of higher-order (lag-0) correlations among neurons, without however providing the identity of the recorded assembly units. A recent *ansatz* by Shimazaki et al. (*Shimazaki et al., 2012*) builds on a state-space model for Poisson point processes developed by Smith and Brown (*Smith and Brown, 2003*) to extract higher-order (lag-0) precise correlation patterns from multiple simultaneously recorded spike trains (see also (*Pipa et al., 2008*; *Gansel and Singer, 2012*; *Picado-Muiño et al., 2013*; *Torre et al., 2013, 2016b*; *Billeh et al., 2014*) for other recent approaches to the detection of groups of synchronous single spikes).

Smith et al. (*Smith and Smith, 2006*; *Smith et al., 2010*) address the problem of testing significance of recurring spike time *sequences* or activity chains like those observed in hippocampal place cells (*Figure 1A, II, IV*; see also [*Abeles and Gerstein, 1988*; *Abeles and Gat, 2001*; *Lee and Wilson, 2004*; *Fujisawa et al., 2008*; *Gerstein et al., 2012*]). Their approach makes use only of the order information in the neural activations, neglecting exact relative timing of spikes or even the number of spikes emitted by each neuron, in order to allow for derivation of exact probabilities based on the multinomial distribution and combinatorial considerations. In a similar vein, Sastry & Unnikrishnan (*Sastry and Unnikrishnan, 2010*) employ data mining techniques like 'market-basket analysis' and the *a-priori*-algorithm to combat the combinatorial explosion problem in sequence detection, scanning first for significant sequential pairs, then based on this subset of pairs for triples, quadruples, and so on, iteratively narrowing down the search space as potential sequences become longer. Several procedures for revealing common modulations in firing rate have been proposed as well (*Peyrache et al., 2010*; *Lopes-dos-Santos et al., 2011*; *Humphries, 2011*; *Lopes-dos-Santos et al., 2013*).

Although some of these techniques are related in one or the other aspect to our algorithm, none of them, to our knowledge, combines all of the features presented here. Our procedure combines fast parametric (bootstrap-free) testing with a fast agglomeration algorithm. This enables to consider, in a heuristic sense, all potential cell combinations with a large range of different temporal lags. We furthermore introduce a novel way for dealing with non-stationarity which is fast and allows to utilize the complete data set for assembly

estimation, rather than slicing it into sufficiently short windows or using computationally demanding bootstrapping. Finally, we showed how a parametric statistic for evaluating the deviation of joint spike distributions from independence can be obtained also for series of counts larger than one by splitting the process into several binary streams. This enables to treat processes developing on slower scales (larger bin widths) within the same statistical framework without loss of information, another novel feature introduced here. The statistical tools developed here may thus allow to readdress important questions about the nature of neural coding in different brain areas, without requiring the researcher to sign up for any particular assembly concept or theoretical framework a priori.

## Comparison with linear decomposition and correlation-based methods

The most popular choices for studying pairwise interactions and (synchronous) multiple-unit structures are, respectively, (Pearson-type) cross-correlations (e.g. [*Shadlen and Newsome, 1998*; *Brody, 1999*]) and principal component analysis (PCA; e.g. [*Peyrache et al., 2010*; *Lopes-dos-Santos et al., 2011*]), owing to their methodological simplicity. In comparison to the methods developed here, there are a number of important issues to note:

First, both standard cross-correlation and PCA are purely *linear* techniques. For strictly binary series, the Pearson cross-correlation is equivalent to computing the deviation of the joint spiking probability from the product of its marginals in the numerator, $p(A,B)-p(A)p(B)$ (e.g., [*Quiroga-Lombard et al., 2013*]). For larger counts or more than two units, however, cross-correlations and PCA, unlike our method, do not capture *nonlinear* interactions or higher-order joint probabilities. Indeed, PCA, strictly, does not even extract correlations among units but rather variance-maximizing directions from the multiple single-unit activity space (e.g. [*Krzanowski, 2000*; *Durstewitz, In press*]), which is a different objective and may lead to results different from methods aimed directly at correlations (e.g., [*Yu et al., 2009*]).

Second, importantly, for either cross-correlation or PCA based methods, statistical significance of the unraveled relationships or structures needs to be properly assessed. In particular, false positives should be avoided. This is in itself a nontrivial topic: Several authors have pointed out in the past that interpretation of cross-correlations may be severely plagued by the presence of (inevitable) non-stationarity (like slow rate covariations or stimulus responses) which may compromise 'traditional' testing by means of, e.g., mere bin or inter-spike-interval shuffling (*Brody, 1999*; *Grün, 2009*; *Quiroga-Lombard et al., 2013*). Indeed, as illustrated in *Appendix 1—figure 1*, superfluous peaks in the cross-correlation may occur, and even flagged up as significant by conventional bootstrapping, although in reality spike times for the two units were drawn independently. Hence, more sophisticated bootstrapping and sliding window analyses have to be afforded that take into account auto-correlations in the time series and non-stationarity (*Davison and Hinkley, 1997*). But these imply that statistical quantities need to be recalculated hundreds to thousands of times, a heavy computational burden that may severely restrict assembly assessment to a limited number of units or few possible lag constellations and temporal scales only (given that this is an NP-hard combinatorial optimization problem [*Nakahara and Amari, 2002*; *Staude et al., 2010*]). In fact, the majority of assembly detection methods focus mostly on synchronous activity (*Wilson and McNaughton, 1994*; *Grün et al., 2003*; *Peyrache et al., 2010*; *Lopes-dos-Santos et al., 2011*). Moreover, these methods usually come with some ad-hoc choices, e.g., the to be used window or block length (in case of block permutations; e.g. [*Efron and Tibshirani, 1993*]), for which there is likely no globally optimal choice across the whole time series. For PCA based methods, sometimes the theoretical Marčenko-Pastur distribution has been used for assessing significance of eigenvalues (*Peyrache et al., 2010*; *Lopes-dos-Santos et al., 2011*), but as illustrated in *Appendix 1—figure 2*, this may be quite misleading especially in the case of non-stationary data.

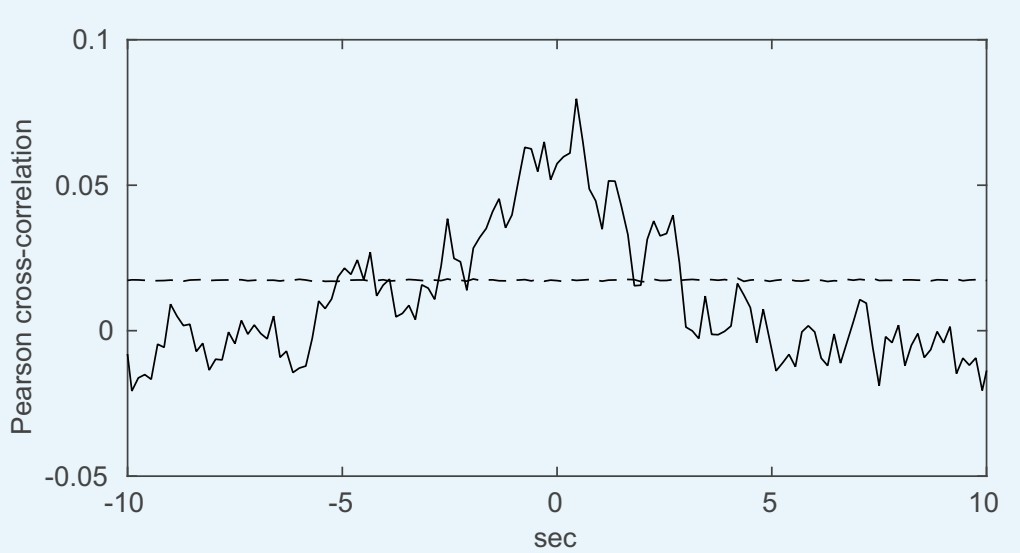

**Appendix 1—figure 1.** Spurious peaks in the cross-correlation function due to non-stationarity. Two units with Poisson spike trains ($T$=1900 s) and step-like rate variations of length $L$=0.5 s (as, e.g., induced by a stimulus) were simulated. The onsets of the rate steps were drawn *independently* for the two units from normal distributions $N(t_i, \sigma^2)$ centered at randomly selected time points $t_i$ (with $\sigma = 2$ s). The Pearson cross-correlation was computed (binning $\Delta = 0.15$ s) and tested for significance using inter-spike-interval shuffling (3000 repetitions). Dashed line indicates 2 standard deviations from mean. For this same simulation setup, our method correctly indicated the absence of true spike time dependencies when applied with the same bin width as used for the cross-correlogram.

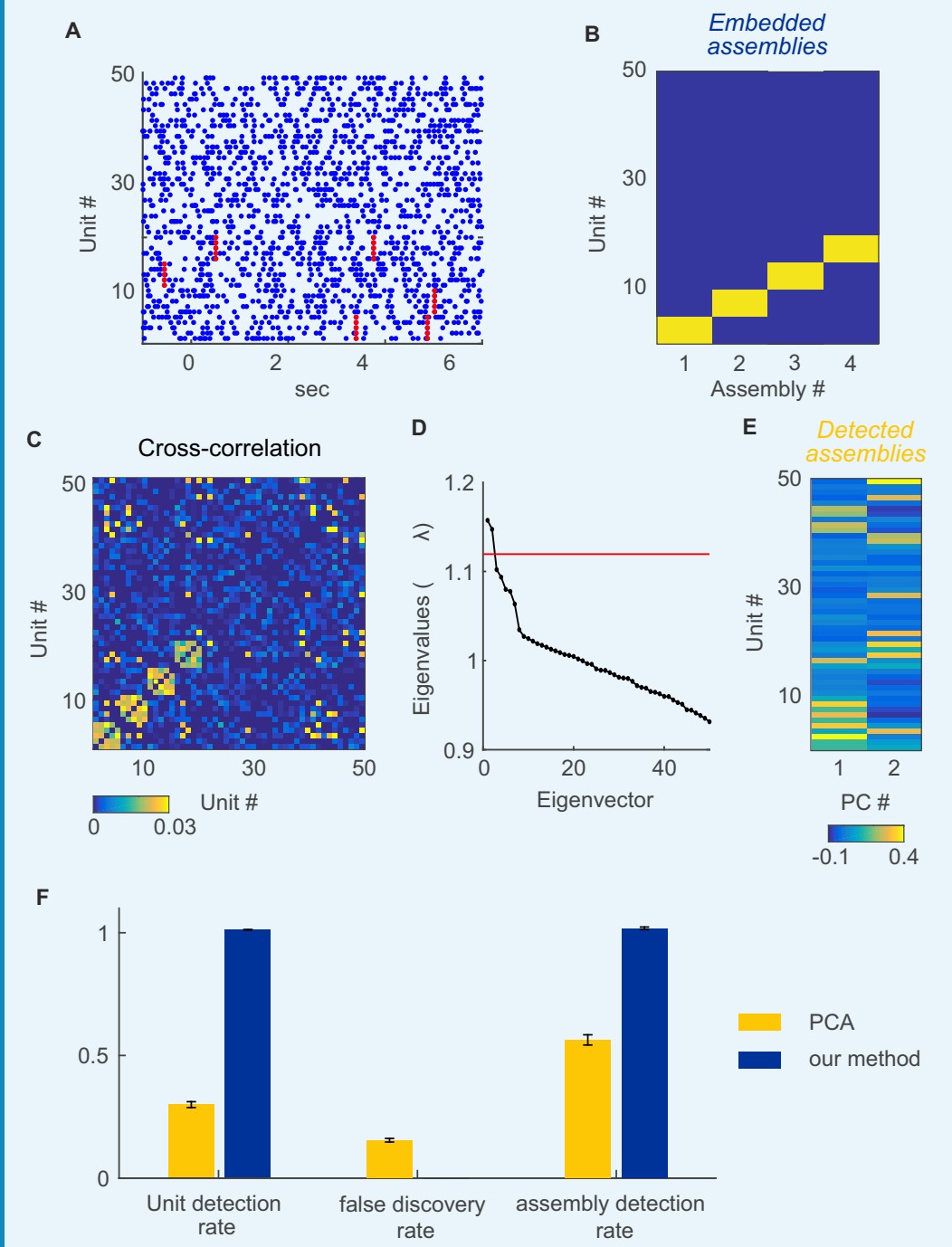

**Appendix 1—figure 2.** Assembly detection with PCA under non-stationarity. For comparison with PCA-based assembly detection methods, simulations were performed with 50 non-stationary Poisson spike trains with four embedded, disjoint assemblies. Assemblies were defined as synchronous spike events (i.e., 'type I', cf. *Figure 1A*; 250 activations in total) occurring at random times within a set of five units. Non-stationary events were implemented as step-like changes shared among 4 groups of 5 units each, randomly chosen from the 50 units simulated, at random timings as described in sect. "*Limitations of parametric testing under non-stationarity*" (parameters used here were Δ=0.02 s, $m$=250, $L$=1 s, $T$=1950 s, baseline rate=5 *Hz*, up-state rate=10 *Hz*). For assembly detection by PCA, based on the cross-correlation matrix indicated in C, we followed the procedure described in Lopes-dos-Santos et al. (Lopes-dos-Santos et al., 2011) using code made publically

available by the authors. (**A**) Examples of spike trains with assembly occurrences marked in red. (**B**) True unit-assembly assignment matrix. (**C**) Cross-correlation matrix with diagonal set to zero for better visualization. (**D**) Eigenvalue spectrum. Red line marks the upper limit of the Marčenko-Pastur distribution. (**E**) 'Loadings' of units on the two only significant principal components, indicating the assignment of units to the two assemblies detected this way. (**F**) Fraction of correctly detected assembly units (left), fraction of units falsely assigned to an assembly (center), and fraction of correctly detected assemblies (right) for PCA (yellow bars) and our method (blue bars). Error bars = SEM, based on 50 independent simulation runs.

Third, cross-correlation analysis still needs to be augmented with some agglomeration scheme that builds up higher-order structures from the pairwise interactions, again a non-trivial endeavor in its own right, especially if various time lag constellations are to be evaluated. PCA, on the other hand, in its basic and most applied formulation only recovers strictly synchronous activity. PCA also comes with a number of other inherent short-comings, (a) because it is not geared toward identifying correlations but, as noted above, variance-maximizing directions (***Krzanowski, 2000***; ***Yu et al., 2009***; ***Durstewitz, In press***), (b) because it may produce assemblies of very unequal size since it places most variation on the first factor, after which variance contributions often fall off exponentially, and (c) because of quite high susceptibility to noise if assembly structures are not very clear-cut (***Appendix 1—figure 3***).

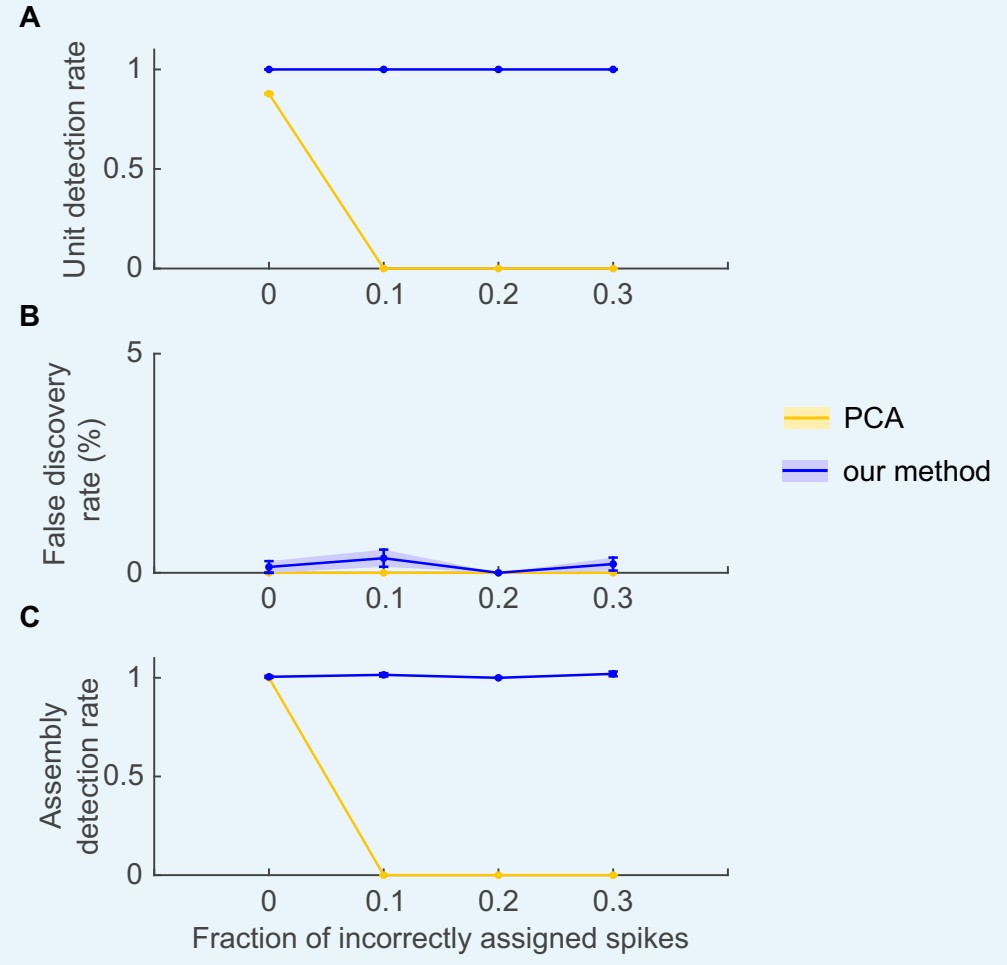

**Appendix 1—figure 3.** Stability of PCA solutions to degradation in assembly patterns. Even under fully stationary conditions, PCA may completely fail to detect assembly patterns if degraded by spike assignment noise. Simulation setup was as in ***Appendix 1—figure 2***, with

the exception that *no* non-stationary step changes were included, i.e. spike trains were completely stationary. PCA methods were implemented as in *Appendix 1—figure 2* (see ref. [*Euston et al., 2007*]). Noise in the form of spike misattributions or spike failures was introduced by randomly removing a fraction of assembly spikes from each spike train. (**A**) Fraction of correctly detected assembly units as a function of the proportion of spike assignment errors, (**B**) fraction of units incorrectly assigned to an assembly (false discovery rate), and (**C**) fraction of correctly detected assemblies (out of the total number of embedded assemblies), for PCA (yellow curves) and our method (blue curves). Error bars = SEM, based on 50 independent simulation runs.

Our algorithm aims to address all these statistical and computational issues in one go, provides proper and fast statistical assessment of unit interactions, unconfounded by non-stationarity within the limits shown (*Figure 7*, *Figure 7—figure supplements 1*,*2*), does this for all possible lag constellations (not just synchrony), and comes with an efficient, statistically anchored algorithm for detecting higher-order structures.

