## [Decision Letter]

Thank you for submitting your article "Cell assemblies at multiple time scales with arbitrary lag distributions" for consideration by *eLife*. Your article has been reviewed by three peer reviewers, one of whom, Marc Howard (Reviewer #1), is a member of our Board of Reviewing Editors and the evaluation has been overseen by David Van Essen as the Senior Editor. The following individuals involved in review of your submission have agreed to reveal their identity: Peter König (Reviewer #2); Bruno Averbeck (Reviewer #3).

The reviewers have discussed the reviews with one another and the Reviewing Editor has drafted this decision to help you prepare a revised submission.

Summary:

Noting the many definitions of “cell assembly” the paper introduces a novel technique for measuring coordinated motifs of sequential firing. This method is computationally efficient and, because it makes minimal assumptions about scale, could be widely used to clarify the notion of cell assemblies, a much-needed theoretical development. Results from different brain regions in different behavioral tasks show a range of time scales suggesting that these reflect a diversity of mechanisms supporting sequences of firing patterns.

Essential revisions:

1) It is essential that a revision better discuss the filtering properties implicit in the method and address a specific concern:

The binning is equivalent to a boxcar average, i.e. temporal filtering with a rectangular window. I could say no window, see Press, W. H., & Teukolsky, S. A. (1988). Numerical recipes in C. Cambridge Figure 2.4.2. This implies that different frequencies are mapped to very different frequency bins. As here no Fourier analysis is performed, in itself it is not that bad. However, the comparison with the time reversed pattern is for different patterns at different distance. Thus, the implicit filtering is suddenly very important. At the very least, the sensitivity varies for patterns of different distance. Moreover, for the important case of perfect 0-lag synchrony, the choice of reference is important. Too close by might lead to a spill over, too far away might allow non stationarities to creep in.

This problem might be addressed either by a convolution with e.g. a Gaussian kernel before binning, in order to create well defined and better behaved temporal properties. And/or, a standard and uniform across tests distance to the reference bin could help. Alternatively, I'm open to very good arguments, why it is not a problem after all jointly with an investigation of sensitivity.

2) A revision should better discuss the generality of the method. The method is advertised as a unified tool for all cases. Although I like it a lot, there might be a few gaps. Consider an avalanche model, where the avalanche might propagate in either one of the other direction. This creates a pair of above chance co-activation with reverse sequence, the present method is blind to. Such limitations have to be better discussed.

---

## [Author Response]

[…]

Essential revisions:

1) It is essential that a revision better discuss the filtering properties implicit in the method and address a specific concern:

The binning is equivalent to a boxcar average, i.e. temporal filtering with a rectangular window. I could say no window, see Press, W. H., & Teukolsky, S. A. (1988). Numerical recipes in C. Cambridge Figure 13.4.2. This implies that different frequencies are mapped to very different frequency bins. As here no Fourier analysis is performed, in itself it is not that bad. However, the comparison with the time reversed pattern is for different patterns at different distance. Thus, the implicit filtering is suddenly very important. At the very least, the sensitivity varies for patterns of different distance.

Yes, we agree, this is a very valid point, and we appreciate the referees' insight on this.

As the referees noted, there are two steps in our method that may imply some sort of temporal filtering. The first is the choice of bin size itself, as larger bins imply stronger low-pass filtering. This is, we think, however, not a problematic, or in fact even desired, feature of our algorithm, since the purpose of evaluating larger bins is precisely to search for structure with coarser granularity, averaging across finer temporal details (which themselves are addressed through the choice of smaller bin widths).

The second type of 'filtering', as the referees indicate, is through the choice of target and reference lag. It may be important to first stress that this ‘filtering’ does not affect the detection of significant patterns at a given lag, other than through the fact that non-stationarity is removed 'only' at time scales defined by this lag (see subsection “Statistical test for pairwise interaction“). This is to say that our significance test will always be valid as long as the process is stationary across at least the length of the lag considered.

However, it may affect the comparisons of patterns at different lags, as essentially for the smaller lags non-stationarity on finer time scales is corrected (in this sense enhancing temporal ‘sensitivity’, as the referees correctly pointed out).

We address this issue in a new section (“Choice of reference (correction) lag”) where we generally advice choosing the reverse lag, for several reasons, unless an explicit comparison of the relevance of patterns spanning different lag sizes is central to the research question. In this case, where this issue of different ‘sensitivities’ cannot be fully avoided, we now recommend choosing lmax+1 (where lmax is the maximum lag tested) as a common reference lag for all target lags considered (thus eliminating differential ‘filtering’ as a possible confound).

Although there are theoretical reasons why sometimes one choice of reference lag may be preferred over the other, additional analyses we have run now on the experimental data sets (also reported in the new section “Choice of reference (correction) lag”) suggest that, in practice, the precise choice of reference lag is not overly crucial and the significant pairs detected by one or the other method largely overlapped (>96%, including the 0-lag case).

Moreover, for the important case of perfect 0-lag synchrony, the choice of reference is important. Too close by might lead to a spill over, too far away might allow non stationarities to creep in.

Yes, this is correct, and the most pragmatic solution to the problem may be to evaluate 0-lag assemblies for a few different reference lags to confirm that the structure unraveled is largely insensitive to this precise choice (this is now included in the revised text as an explicit recommendation in the Discussion section and *Materials and methods* section).

We have evaluated this issue now in more depth by probing the detection of synchronous assemblies at both fine (‘type I’, cf. Figure 1) and broad (‘type V’) time scales with a set of different reference lags (new Figure 1—figure supplement 1). While we observed that indeed using too small a reference lag (l*=−1) starts losing some structure while too large lags (l*<−3) diminish temporal sensitivity, generally these effects were rather mild in the sense that detection of 0-lag structure was, once again (see above), rather robust and independent of the precise choice of reference lag. Thus, from these theoretical and empirical considerations, one may derive the recommendation to use l*=−2 as a default for the synchronous case, as in the present manuscript, to safeguard against potential spillover issues, but possibly probe with a few different reference lags if unsure about this particular choice for a given experimental situation.

This problem might be addressed either by a convolution with e.g. a Gaussian kernel before binning, in order to create well defined and better behaved temporal properties. And/or, a standard and uniform across tests distance to the reference bin could help. Alternatively, I'm open to very good arguments, why it is not a problem after all jointly with an investigation of sensitivity.

We have addressed this issue as indicated in our previous replies above, that is, by basically following the referees' second and third recommendation, providing both some theoretical arguments for the choices made, but also rerunning analyses with a common reference lag, as suggested, and performing some sensitivity analyses for the lag-0 case (reported in the new sect. “Choice of reference (correction) lag”).

Convolution with a Gaussian kernel is a nice idea which we have frequently relied on in the past (e.g., Durstewitz et al. 2010, Lapish et al., 2015, J Neurosci, 35(28):10172–10187), but is less straightforward to apply in the present case since, strictly, our test statistic is derived for count (integer-valued) variables (although extensions are certainly possible).

2) A revision should better discuss the generality of the method. The method is advertised as a unified tool for all cases. Although I like it a lot, there might be a few gaps. Consider an avalanche model, where the avalanche might propagate in either one of the other direction. This creates a pair of above chance co-activation with reverse sequence, the present method is blind to. Such limitations have to be better discussed.

We thank the referees for bringing this up. Sequences repeated in reverse direction are indeed an important special case (please note, however, that avalanches, in particular, at least by some authors have been attributed to feedforward networks like synfire chains (e.g., Beggs and Plenz, 2003), which, if we are not mistaken, would imply directionality?). In the revised version, we have now included this point about reverse sequences, and have extended it into a wider discussion about the choice of reference lag and its implications and limitations (sect. ‘Limitations of parametric testing under non-stationarity’, new subsection “Choice of reference (correction) lag”). We have also included a new paragraph on limitations associated with the proposed difference statistic and choice of reference lag into the main Discussion section.

For the specific case of reverse sequences, we note that another choice of reference lag may provide a remedy, as discussed now in subsection “Choice of reference (correction lag).